# Heterozygous expression of a *Kcnt1* gain-of-function variant has differential effects on somatostatin- and parvalbumin-expressing cortical GABAergic neurons

Amy N Shore[1,2], Keyong Li[1], Mona Safari[1,3], Alshaima'a M Qunies[4,5], Brittany D Spitznagel[6,7,8], C David Weaver[6,7,8], Kyle Emmitte[4], Wayne Frankel[9,10], Matthew C Weston[1,2,3,11]*

[1]Fralin Biomedical Research Institute at Virginia Tech Carilion, Center for Neurobiology Research, Roanoke, United States; [2]Department of Neurological Sciences, University of Vermont, Burlington, United States; [3]Translational Biology, Medicine, and Health Graduate Program, Fralin Biomedical Research Institute at Virginia Tech Carilion, Roanoke, United States; [4]Department of Pharmaceutical Sciences, UNT System College of Pharmacy, University of North Texas Health Science Center, Fort Worth, United States; [5]School of Biomedical Sciences, University of North Texas Health Science Center, Fort Worth, United States; [6]Department of Pharmacology, Vanderbilt University, Nashville, United States; [7]Vanderbilt Institute of Chemical Biology, Vanderbilt University, Nashville, United States; [8]Department of Chemistry, Vanderbilt University, Nashville, United States; [9]Institute for Genomic Medicine, Columbia University, New York, United States; [10]Department of Neurology, Columbia University, New York, United States; [11]School of Neuroscience, Virginia Tech, Blacksburg, United States

*For correspondence:
mcweston7c@vt.edu

^other^**Conflict of interest statement:** The authors declare no competing financial interests.

**Abstract** More than 20 recurrent missense gain-of-function (GOF) mutations have been identified in the sodium-activated potassium ($K_{Na}$) channel gene *KCNT1* in patients with severe developmental and epileptic encephalopathies (DEEs), most of which are resistant to current therapies. Defining the neuron types most vulnerable to KCNT1 GOF will advance our understanding of disease mechanisms and provide refined targets for precision therapy efforts. Here, we assessed the effects of heterozygous expression of a *Kcnt1* GOF variant (*Kcnt1*^Y777H^) on $K_{Na}$ currents and neuronal physiology among cortical glutamatergic and GABAergic neurons in mice, including those expressing vasoactive intestinal polypeptide (VIP), somatostatin (SST), and parvalbumin (PV), to identify and model the pathogenic mechanisms of autosomal dominant *KCNT1* GOF variants in DEEs. Although the *Kcnt1*^Y777H^ variant had no effects on glutamatergic or VIP neuron function, it increased subthreshold $K_{Na}$ currents in both SST and PV neurons but with opposite effects on neuronal output; SST neurons became hypoexcitable with a higher rheobase current and lower action potential (AP) firing frequency, whereas PV neurons became hyperexcitable with a lower rheobase current and higher AP firing frequency. Further neurophysiological and computational modeling experiments showed that the differential effects of the *Kcnt1*^Y777H^ variant on SST and PV neurons are not likely due to inherent differences in these neuron types, but to an increased persistent sodium current in PV, but not SST, neurons. The *Kcnt1*^Y777H^ variant also increased excitatory input onto, and chemical and electrical synaptic connectivity between, SST neurons. Together, these data suggest differential pathogenic mechanisms, both direct and compensatory, contribute to disease phenotypes, and provide a salient example of how a pathogenic ion channel variant can

cause opposite functional effects in closely related neuron subtypes due to interactions with other ionic conductances.

## eLife assessment

Shore et al. report the **important** effects of a heterozygous mutation in the KCNT1 potassium channel on ion currents and firing behavior of excitatory and inhibitory neurons in the cortex of KCNT1-Y777H mice. The authors provide **solid** evidence of physiological differences between this heterozygous mutation and their previous work with homozygotes. The reviewers appreciated the inclusion of recordings in ex vivo slices and dissociated cortical neurons, as well as the additional evidence showing an increase in persistent sodium currents in parvalbumin-positive interneurons in heterozygotes.

## Introduction

Heterozygous missense variants have been identified in the sodium-gated potassium channel gene *KCNT1* in more than 200 individuals exhibiting a wide spectrum of developmental and epileptic encephalopathies (DEEs), with the majority being classified as either epilepsy of infancy with migrating focal seizures or autosomal dominant or sporadic sleep-related hypermotor epilepsy (*Barcia et al., 2012*; *Bonardi et al., 2021*; *Heron et al., 2012*). Each of these epilepsy syndromes result in early-onset, frequent seizures that are largely pharmacoresistant and often accompanied by a range of cognitive, psychiatric, and motor impairments. Thus, there is a critical need for a better understanding of how heterozygous expression of these *KCNT1* variants in the developing brain alters neuronal physiology and network behavior to lead to such devastating neurodevelopmental disorders.

*KCNT1* encodes a tetrameric potassium channel that is widely expressed in both glutamatergic and GABAergic neurons of the brain, particularly those of the cerebellum, striatum, thalamus, hippocampus, and cortex (*Berg et al., 2007*; *Bhattacharjee et al., 2002*; *Gertler et al., 2022*; *Rizzi et al., 2016*). Although its precise role in normal physiology is not well understood, at least in some neuronal types, KCNT1 is activated by a persistent inward sodium leak ($Na_P$) current at rest, where it has a proposed role in fine tuning neuronal excitability by countering the effects of the $Na_P$ current across subthreshold voltages (*Budelli et al., 2009*; *Hage and Salkoff, 2012*). Consistent with this role, loss-of-function (LOF) studies using mouse models lacking KCNT1, and the associated sodium-activated potassium ($K_{Na}$) current, have shown enhanced action potential (AP) firing across multiple neuron types (*Evely et al., 2017*; *Liu et al., 2022*; *Lu et al., 2015*; *Martinez-Espinosa et al., 2015*; *Reijntjes et al., 2019*; *Zhang et al., 2022*). Characterizations of pathogenic DEE-associated *KCNT1* variants in heterologous cells found that nearly all cause gain-of-function (GOF) effects on the channel, increasing the associated $K_{Na}$ current (*Hinckley et al., 2023*; *Kim et al., 2014*; *McTague et al., 2018*; *Milligan et al., 2014*; *Tang et al., 2016*). Based on LOF studies, this would be expected to reduce neuronal excitability; however, it is difficult to predict the effects of these GOF variants on AP generation in neurons, particularly among neuronal subtypes, a priori.

To address this knowledge gap, we previously generated and characterized a mouse model expressing a human ADNFLE-associated *KCNT1* GOF variant (Y796H or Y777H in mice) (*Shore et al., 2020*). Although heterozygous expression of the *KCNT1*-Y796H variant is sufficient to cause severe childhood epilepsy in humans, we only observed rare behavioral seizures in heterozygous $Kcnt1^{Y777H}$ mice see Discussion (*Shore et al., 2020*); however, we identified hyperexcitable, hypersynchronous cortical networks and frequent, early-onset seizures in homozygous $Kcnt1^{Y777H}$ mice. As a potential underlying mechanism of these network alterations, we demonstrated that homozygous $Kcnt1^{Y777H}$ expression increases subthreshold $K_{Na}$ currents and reduces excitability in GABAergic neuron populations, particularly in those classified as non-fast spiking (NFS), but it does not alter glutamatergic neuron excitability. We further observed evidence of homeostatic compensation and network remodeling downstream of $K_{Na}$ current increases during development, including increased excitatory input onto glutamatergic and NFS GABAergic neurons, and enhanced homotypic synaptic connectivity. Although these findings provide a strong mechanistic basis for understanding how KCNT1 GOF disrupts neuronal physiology and network behavior to lead to seizure disorders, key issues remain unresolved. First, considering the heterozygous nature of *KCNT1* GOF variants in the overwhelming

majority of *KCNT1*-related DEE patients, it is crucial to determine whether heterozygous *Kcnt1*[Y777H] expression results in similar neuronal impairments and network alterations to those with homozygous expression. Second, more recent studies, using both in silico modeling and additional construct-valid mouse models, have similarly identified impairments in GABAergic neuron excitability downstream of KCNT1 GOF (*Gertler et al., 2022*; *Kuchenbuch et al., 2021*; *Wu et al., 2024*), indicating impaired inhibition as a shared pathogenic mechanism in *KCNT1*-related DEEs; however, precisely which GABAergic subtypes are most impacted, and how, remains unknown.

Here, we assessed the effects of heterozygous *Kcnt1*[Y777H] expression on $K_{Na}$ currents and neuronal physiology among cortical glutamatergic and GABAergic neurons, including those expressing vaso-active intestinal polypeptide (VIP), somatostatin (SST), and parvalbumin (PV). Initial assessments of cortical neuron populations with heterozygous *Kcnt1*[Y777H] expression showed strikingly similar effects on $K_{Na}$ currents and AP generation to those with homozygous expression, although not surprisingly, these effects were lesser in magnitude. Across all cortical neuron types examined, the heterozygous *Kcnt1*[Y777H] variant caused a range of effects on neuronal excitability and AP generation, from no change (glutamatergic and VIP GABAergic) to decreased excitability (SST GABAergic) to increased excitability (PV GABAergic). Neuron types that showed no change had $K_{Na}$ currents that were only significantly increased at suprathreshold voltages and exhibited a steeper voltage dependence of activation. Interestingly, both SST and PV neurons showed similar increases in $K_{Na}$ currents across subthreshold voltages, however, only PV neurons had additional increases in the persistent $Na^+$ current, which modeling experiments indicated was sufficient to overcome the effects of KCNT1 GOF and cause an overall increase in AP generation. SST neurons also showed an increase in excitatory input, and in homotypic electrical and chemical coupling. Taken together, these data provide further evidence of the enhanced vulnerability of GABAergic neurons, particularly those expressing SST and PV, to KCNT1 GOF. Moreover, these data show that heterozygous expression of a single *KCNT1* GOF variant can result in a complex array of neuron-type-dependent effects, both direct and indirect, each potential contributor to the neural circuit pathology underlying *KCNT1*-related DEEs.

## Results

### Heterozygous *Kcnt1*[Y777H] expression alters the shape and frequency of APs in NFS GABAergic neurons

We previously identified frequent, early-onset seizures in mice with homozygous expression of the *Kcnt1* GOF variant Y777H (hereafter referred to as YH-HOM), and as a potential underlying pathological mechanism, we demonstrated that homozygous *Kcnt1*[Y777H] expression drastically impairs AP shape and generation in NFS GABAergic cortical neurons, with lesser effects on fast spiking (FS) GABAergic, and no significant effects on glutamatergic, cortical neurons (*Shore et al., 2020*). Considering that patients with *KCNT1*-associated epilepsy are predominantly heterozygous for *KCNT1* GOF variants, it is crucial to determine whether heterozygous expression of these variants, which likely leads to the formation of heteromeric channels consisting of wildtype (WT) and mutant subunits, results in similar neuronal impairments. To assess neuron subtype-specific effects of heterozygous *KCNT1* GOF on membrane properties and AP firing, we isolated and cultured cortical neurons from pups with heterozygous *Kcnt1*[Y777H] expression (hereafter referred to as YH-HET), and their WT littermates, at postnatal day 0 (P0). After infecting the cultured neurons with AAV-*CaMKIIa*-GFP to facilitate glutamatergic neuron identification, we performed whole-cell, current-clamp analysis between 13 and 17 days in vitro (DIV). Moreover, to compare to homozygous KCNT1 GOF effects observed previously, the recorded neurons were classified as glutamatergic, FS GABAergic, or NFS GABAergic, based on GFP expression, AP parameters, and evoked synaptic responses (see Methods).

Current-clamp recordings from YH-HET and WT glutamatergic neurons showed no significant differences in any membrane or AP shape property measured (*Figure 1A₁* and *Supplementary file 1*), similar to observations from the homozygous KCNT1 GOF studies. Accordingly, AP firing frequencies across increasing current steps in YH-HET glutamatergic neurons were not altered compared with those of WT (*Figure 1B₁*). For FS GABAergic neurons, we previously showed that homozygous *Kcnt1*[Y777H] expression increases the rheobase—the minimal amount of current necessary to induce an AP—and reduces the AP firing frequency. Heterozygous *Kcnt1*[Y777H] expression did not alter the rheobase, or any other passive or active membrane property of FS GABAergic neurons (*Figure 1A₂*

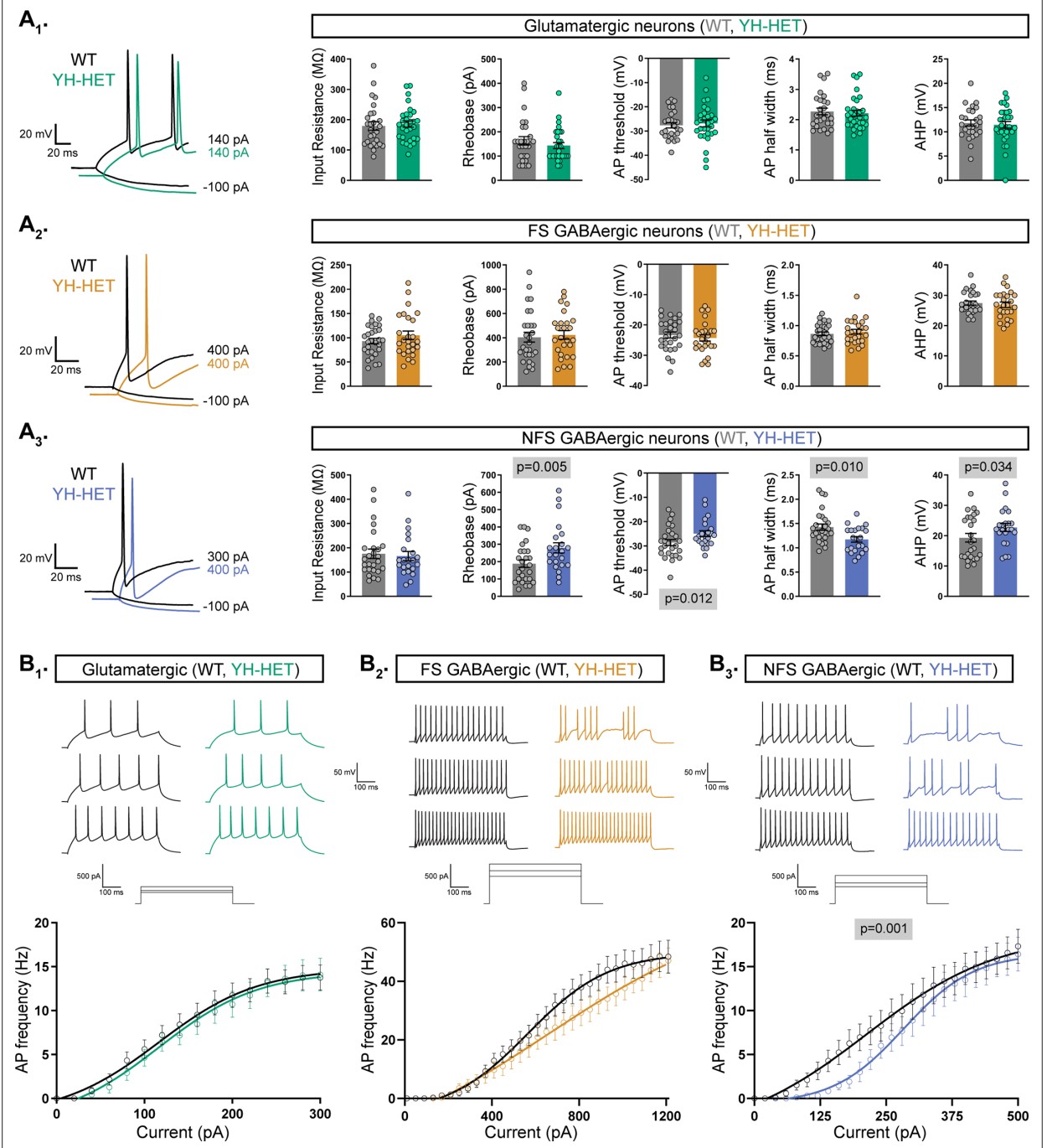

**Figure 1.** Heterozygous *Kcnt1*^Y777H^ expression alters action potential (AP) shape and generation in non-fast spiking (NFS) GABAergic neurons. (**A₁–A₃**) On the left, representative responses to step currents are shown for glutamatergic, and fast spiking (FS) and NFS GABAergic, wildtype (WT) (black) and YH-HET (colors) neurons (top to bottom), illustrating the input resistance (in response to a hyperpolarizing step) and the rheobase (the first trace with an AP in response to a depolarizing step) for each neuron type. On the right, bar graphs show quantification and mean ± standard error of the mean (SEM) of the membrane properties and AP parameters for each neuron type for WT (gray) and YH-HET (colors) groups, with individual neuron measurements overlaid in scatter plots. The p-values are shown on each graph where p < 0.05. (**B₁–B₃**) Representative traces are shown at low, medium, and high current steps for glutamatergic, and FS and NFS GABAergic, WT (black) and YH-HET (colors) neurons (left to right). The line graphs below show the number of APs (mean ± SEM) per current injection step in WT (black) and YH-HET (colors) neurons. Statistical significance was tested using generalized linear mixed models, and p-values are shown on each graph where p < 0.05. Data from this figure are from 9 WT and 9 YH-HET littermate pups, with the following neuron sample sizes: glutamatergic WT (n=27) and YH-HET (n=30), FS GABAergic WT (n=27) and YH-HET (n=29), and NFS GABAergic WT (n=26) and YH-HET (n=22).

*Figure 1 continued on next page*

*Figure 1 continued*

The online version of this article includes the following source data for figure 1:

**Source data 1.** File containing data used to generate the graphs in *Figure 1*.

and *Supplementary file 1*). Although YH-HET FS GABAergic neurons appeared to fire fewer APs than their WT counterparts, particularly at higher current steps, this effect was not significant (*Figure 1B$_2$*). Lastly, we previously showed that homozygous *Kcnt1*$^{Y777H}$ expression has the strongest effects in NFS GABAergic neurons, showing a decrease in input resistance, accompanied by an increase in rheobase and a reduction in AP firing frequency. YH-HOM NFS GABAergic neurons also have narrower AP half-widths and larger afterhyperpolarizations (AHPs) than those of WT. Although heterozygous *Kcnt1*$^{Y777H}$ expression in NFS GABAergic neurons did not cause a significant decrease in input resistance (WT: 171 ± 16; YH-HET: 146 ± 14, p = 0.19), it did increase the rheobase and reduce the AP firing frequency, particularly at lower current steps, relative to those of WT (*Figure 1A$_3$, B$_3$*). YH-HET NFS GABAergic neurons also had narrower APs, larger AHPs, and more depolarized AP thresholds than those of WT (*Supplementary file 1*). Together, these data demonstrate that the neurophysiological effects on cortical neurons with monoallelic expression of the *Kcnt1*$^{Y777H}$ variant, expressing channels with mutant and WT subunits, are similar to those with biallelic expression, expressing only mutant subunits, with both causing the strongest impairments in NFS GABAergic neurons.

## Heterozygous *Kcnt1*$^{Y777H}$ expression increases subthreshold K$_{Na}$ currents in NFS GABAergic neurons

Previously, we showed that homozygous *Kcnt1*$^{Y777H}$ expression in GABAergic cortical neurons increases the K$_{Na}$ current across subthreshold voltages, an effect that is particularly evident in NFS GABAergic neurons; conversely, in glutamatergic cortical neurons with homozygous expression of the same variant, increases in K$_{Na}$ currents are only apparent at depolarized voltages (>+30 mV) (*Shore et al., 2020*). To assess the effects of heterozygous *Kcnt1*$^{Y777H}$ expression on KCNT1 channel function, we measured the associated K$_{Na}$ current in each cortical neuron subtype. We recorded K$_{Na}$ currents by applying voltage steps to voltage-clamped neurons and comparing the delayed outward current before and after the addition of the voltage-gated sodium channel inhibitor tetrodotoxin (TTX) (*Figure 2A$_{1-3}$*). As reported previously, there were K$_{Na}$ currents in all three WT neuron subtypes, beginning around −10 mV and increasing with depolarization (*Figure 2A$_{1-3}$, B$_{1-3}$*), whereas at more negative potentials, the TTX-sensitive current was net inward (*Figure 2C$_{1-3}$*) due to the counteracting persistent Na$^+$ current.

In glutamatergic and NFS GABAergic YH-HET neurons, the overall K$_{Na}$ current was increased relative to those of WT, as measured by a significant effect of genotype using a linear model (*Figure 2B$_{1, 3}$*). Importantly, in each of these neuron subtypes, heterozygous *Kcnt1*$^{Y777H}$ expression increased K$_{Na}$ currents with distinct, voltage-dependent patterns that appeared strikingly similar to those reported with homozygous expression. For instance, we previously showed that, across negative potentials, YH-HOM and WT glutamatergic neuron K$_{Na}$ currents are indistinguishable, whereas at more positive potentials (>+30 mV), the *Kcnt1*$^{Y777H}$ variant causes significant increases in K$_{Na}$ currents. Similar voltage-dependent effects were observed in YH-HET glutamatergic neurons (*Figure 2B$_1$, C$_1$*), although pairwise comparisons showed that K$_{Na}$ current increases at positive potentials in the YH-HET glutamatergic neurons were not significant. Conversely, we previously demonstrated broad increases in K$_{Na}$ currents across negative potentials, with significant increases from −60 to +10 mV, in YH-HOM NFS GABAergic neurons compared with those of WT, with lesser effects across positive potentials. Similar voltage-dependent increases were observed in YH-HET NFS GABAergic neurons, with pairwise comparisons showing significant K$_{Na}$ current increases at −70, −60, −50, and −20 mV (*Figure 2B$_3$, C$_3$*). For both glutamatergic and NFS GABAergic neurons, the magnitudes of the K$_{Na}$ current increases in YH-HET neurons were intermediate to those of WT and YH-HOM neurons (*Figure 2—figure supplement 1*), demonstrating a gene dose-dependent effect of the *Kcnt1*$^{Y777H}$ variant on K$_{Na}$ current increases and validating a GOF effect of the heterozygous *Kcnt1*$^{Y777H}$ variant on channel function. Lastly, although previous studies showed that homozygous *Kcnt1*$^{Y777H}$ expression increases K$_{Na}$ current at several negative voltage steps (−50, −40, and −10 mV) in FS GABAergic neurons, heterozygous expression of the

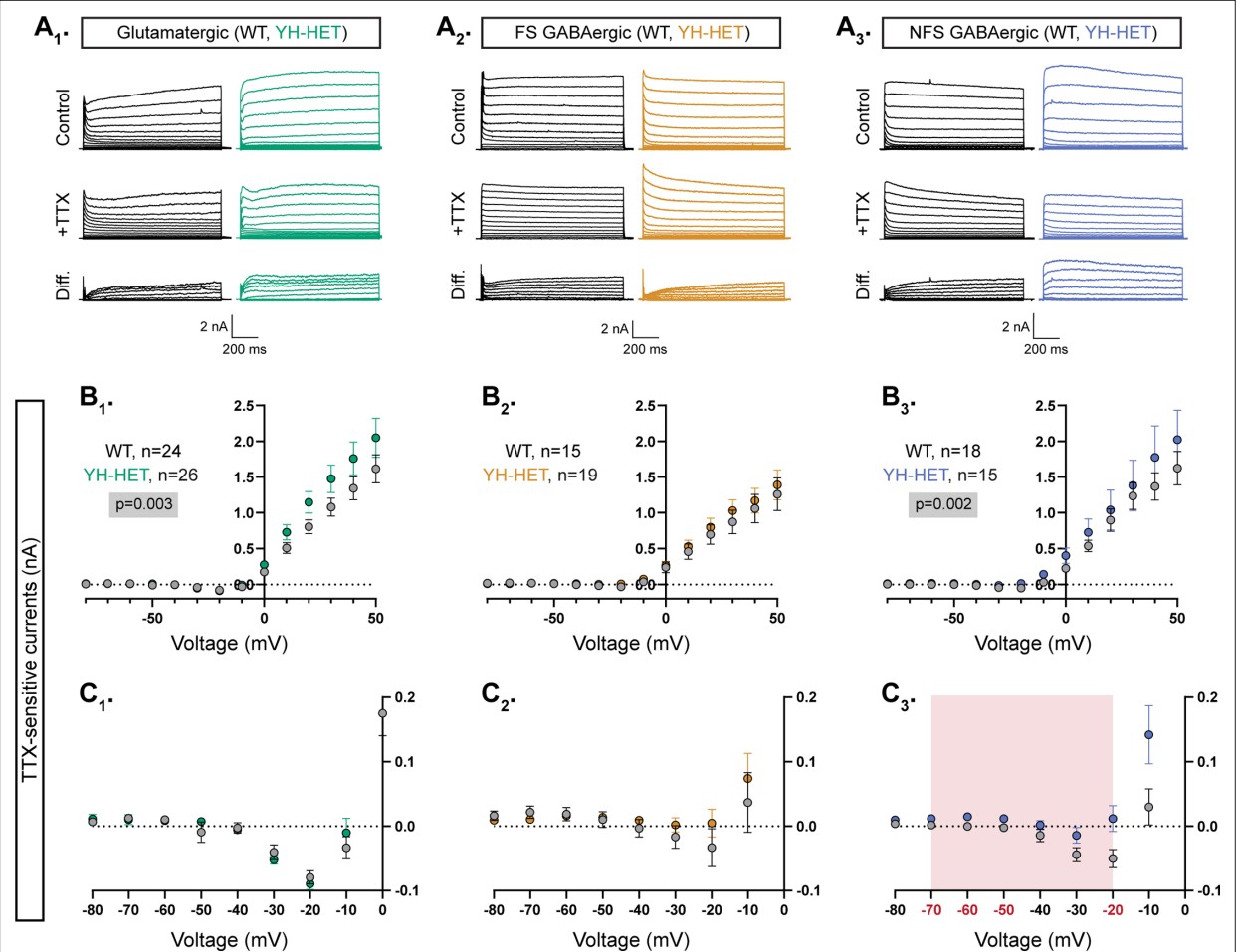

**Figure 2.** Heterozygous *Kcnt1*[Y777H] expression increases subthreshold K_Na currents in non-fast spiking (NFS) GABAergic neurons. (**A₁–A₃**) Representative traces in control (top), 0.5 μM tetrodotoxin (TTX) (middle), and the difference current (bottom) calculated by subtracting the membrane current response to voltage steps (−80 to +50 mV) from a holding potential of −70 mV in TTX from the response in control external solution in glutamatergic, and fast spiking (FS) and NFS GABAergic, wildtype (WT) (black) and YH-HET (colors) neurons. To include all of the representative traces in the figures, and prevent overlap of the traces, the large inward sodium currents were removed from each set of traces using the masking tool in Adobe Illustrator. (**B₁–B₃**) Summary data show the K_Na current (mean ± SEM) for each voltage step in glutamatergic, and FS and NFS GABAergic, WT (black and gray) and YH-HET (colors) neurons. The p-values are shown on each graph where p < 0.05, and the *n* values are the number of neurons recorded for each group. (**C₁–C₃**) Plots of the K_Na current (mean ± SEM) for each voltage step from −80 to 0 mV in WT (black and gray) and YH-HET (colors) neurons to illustrate the values that are too small to be seen on the graphs in B₁–B₃. The shaded red area in C₃ indicates the subthreshold voltage range with significantly higher K_Na currents (red voltages along *x*-axis indicate steps where p < 0.05) in YH-HET relative to WT neurons. Statistical significance for *I–V* plots was tested using generalized linear mixed models with genotype and voltage step as fixed effects followed by pairwise comparisons at each level.

The online version of this article includes the following source data and figure supplement(s) for figure 2:

**Source data 1.** File containing data used to generate the graphs in *Figure 2*.

**Figure supplement 1.** The magnitudes of the K_Na current increases in YH-HET glutamatergic and non-fast spiking (NFS) GABAergic neurons are intermediate to those of wildtype (WT) and YH-HOM neurons.

**Figure supplement 1—source data 1.** File containing data used to generate the graphs in *Figure 2—figure supplement 1*.

same variant in FS GABAergic neurons caused no significant differences in K_Na currents compared with those of WT (*Figure 2B₂, C₂*).

## Heterozygous *Kcnt1*[Y777H] expression does not alter synaptic connectivity or the excitation–inhibition balance

In addition to alterations in the intrinsic passive and active membrane properties of cortical neurons with homozygous *Kcnt1*[Y777H] expression, we previously showed evidence of altered synaptic

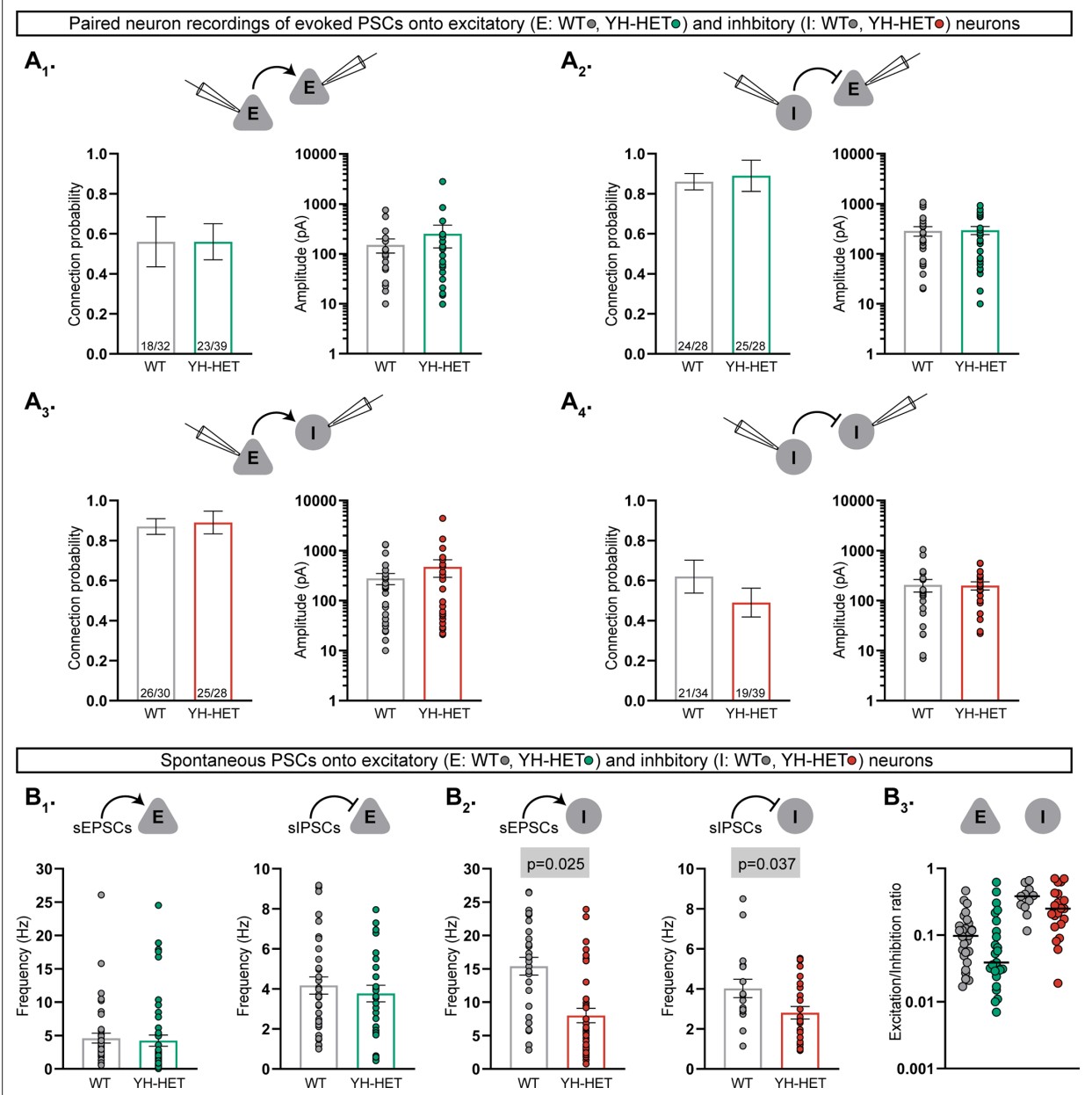

**Figure 3.** Heterozygous *Kcnt1*[Y777H] expression does not alter synaptic connectivity or the excitation–inhibition balance. (**A₁–A₄**) Evoked postsynaptic currents (PSCs) were recorded from neuron pairs glutamatergic (excitatory, E) and GABAergic (inhibitory, I) by stimulating the neuron type indicated on the left and recording the response in the neuron type indicated on the right (wildtype [WT], gray; YH-HET E, green; YH-HET I, red). Bar graphs below each recorded neuron pair schematic show summary data (mean ± SEM) of the connection probability (left graph; numbers on bars represent connected pair number/recorded pair number) and peak evoked PSC amplitude (right graph; dots represent individual evoked responses) between each motif. (**B₁**) Bar graphs with overlaid individual neuron measurements and mean ± SEM show the spontaneous EPSC (sEPSC) or IPSC (sIPSC) frequency onto E neurons (WT, gray; YH-HET, green). (**B₂**) Bar graphs with overlaid individual neuron measurements and mean ± SEM show the sEPSC or sIPSC frequency onto I neurons (WT, gray; YH-HET, red). (**B₃**) Scatter plots show individual E/I ratio measurements onto E neurons (WT, gray; YH-HET, green) and I neurons (WT, gray; YH-HET, red). The p-values are shown on each graph where p < 0.05.

The online version of this article includes the following source data for figure 3:

**Source data 1.** File containing data used to generate the graphs in *Figure 3*.

connectivity and activity, leading to hypersynchronous, hyperexcitable YH-HOM networks; more specifically, we found an increase in homotypic coupling between glutamatergic [excitatory–excitatory (E–E)] and GABAergic [inhibitory–inhibitory (I–I)] neuron pairs, and an increase in the frequency of spontaneous excitatory postsynaptic currents (sEPSCs), accompanied by an increase in the E/I ratio, onto YH-HOM glutamatergic cortical neurons (*Shore et al., 2020*). First, to determine whether there are similar changes in synaptic connectivity in YH-HET networks, we performed paired recordings of glutamatergic (excitatory, E) and GABAergic (inhibitory, I) neurons and alternatively stimulated each neuron at 0.1 Hz to test baseline connection probability and strength at the four possible motifs (E–E, I–E, E–I, and I–I). Connection probability was not altered in YH-HET networks at any of the motifs tested (*Figure 3A₁₋₄*, left bar graphs), and the amplitudes of the evoked postsynaptic currents (ePSCs) between connected neurons were not different between genotypes for any of the four connection types (*Figure 3A₁₋₄*, right bar graphs), indicating grossly normal synaptic interactions among glutamatergic and GABAergic neurons in YH-HET networks. Next, to assess potential alterations in synaptic activity, we recorded spontaneous postsynaptic currents (sEPSCs and sIPSCs) onto voltage-clamped glutamatergic and GABAergic neurons. Unlike the observation of an increase in sEPSC frequency onto YH-HOM glutamatergic neurons, there was no difference in sEPSC, or sIPSC, frequency onto YH-HET glutamatergic neurons (*Figure 3B₁*). Furthermore, although we previously found no alterations in sPSC frequency onto YH-HOM GABAergic neurons, sEPSC and sIPSC frequencies were both slightly reduced onto YH-HET GABAergic neurons (*Figure 3B₂*). Finally, to assess the net effect of altered sPSC activity onto YH-HET neurons, we calculated the E/I ratio, based on the relative frequency and size of the sPSCs, and found no difference in the E/I ratio onto either YH-HET neuron type (*Figure 3B₃*). Thus, although heterozygous *Kcnt1*^Y777H^ expression resulted in similar $K_{Na}$ current increases and neuronal physiology effects to those found with homozygous *Kcnt1*^Y777H^ expression, the broad effects on synaptic connectivity and activity found in YH-HOM networks were absent in YH-HET networks, which likely results in the observed reduction in seizure incidence in heterozygous, relative to homozygous, mice.

## Heterozygous *Kcnt1*^Y777H^ expression differentially affects the intrinsic excitability of SST- and PV-expressing GABAergic neurons

Next, we sought to determine which GABAergic subtypes are most impacted by heterozygous *Kcnt1*^Y777H^ expression. About 80–90% of cortical GABAergic neurons fall into three, largely non-overlapping populations that can be identified by their expression of unique markers: the $Ca^{2+}$-binding protein PV (~40%), the neuropeptides SST (~30%), and VIP (~15%) (*Rudy et al., 2011*; *Tremblay et al., 2016*). The majority of PV-expressing neurons have been characterized as FS, and VIP-expressing as NFS, whereas SST-expressing neurons, although largely thought to show NFS firing properties, also contain a population that exhibits an FS-like phenotype (*Large et al., 2016*; *Ma et al., 2006*). KCNT1 is expressed at higher levels in both human and mouse cortical GABAergic neurons expressing PV and SST, than in those expressing VIP (*Gertler et al., 2022*; *Shore et al., 2020*). Based on the expression profile of KCNT1, and our findings that homozygous and heterozygous expression of the *Kcnt1* GOF variant strongly reduced the excitability of GABAergic neurons with NFS firing properties, we hypothesized that the SST-expressing neurons are the most vulnerable to the effects of KCNT1 GOF. To test this hypothesis, we crossed PV-, SST-, and VIP-Cre mouse lines to the *Kcnt1*^Y777H^ mouse line and cultured neurons from the cortices of WT and YH-HET littermate progeny (*Figure 4A* and *Figure 4— figure supplement 1A*). We infected the cultured neurons with AAV-*CaMKIIa*-GFP, to mark glutamatergic neurons, and AAV-*hSyn*-DIO-mCherry, to mark Cre recombinase-expressing GABAergic neurons, and performed whole-cell, patch-clamp electrophysiology at DIV 13–17 on GFP⁻/mCherry⁺ neurons from each group.

First, we measured the intrinsic membrane properties and AP shape parameters of the three GABAergic subtypes from the WT control groups to verify that they accurately reflected electrophysiological behaviors of these subtypes from previous ex vivo recordings (*Taniguchi et al., 2011*). As expected, the VIP neurons showed a relatively large input resistance, small rheobase, wide AP half-width, small AHP, and low firing rate (*Figure 4—figure supplement 1B, C*, purple); in contrast, PV neurons showed a small input resistance, large rheobase, narrow AP half-width, large AHP, and high firing rate (*Figure 4—figure supplement 1B, C*, orange). For SST neurons, the values measured for input resistance, rheobase, AP half-width, AHP, and firing rate were all intermediate to those of VIP

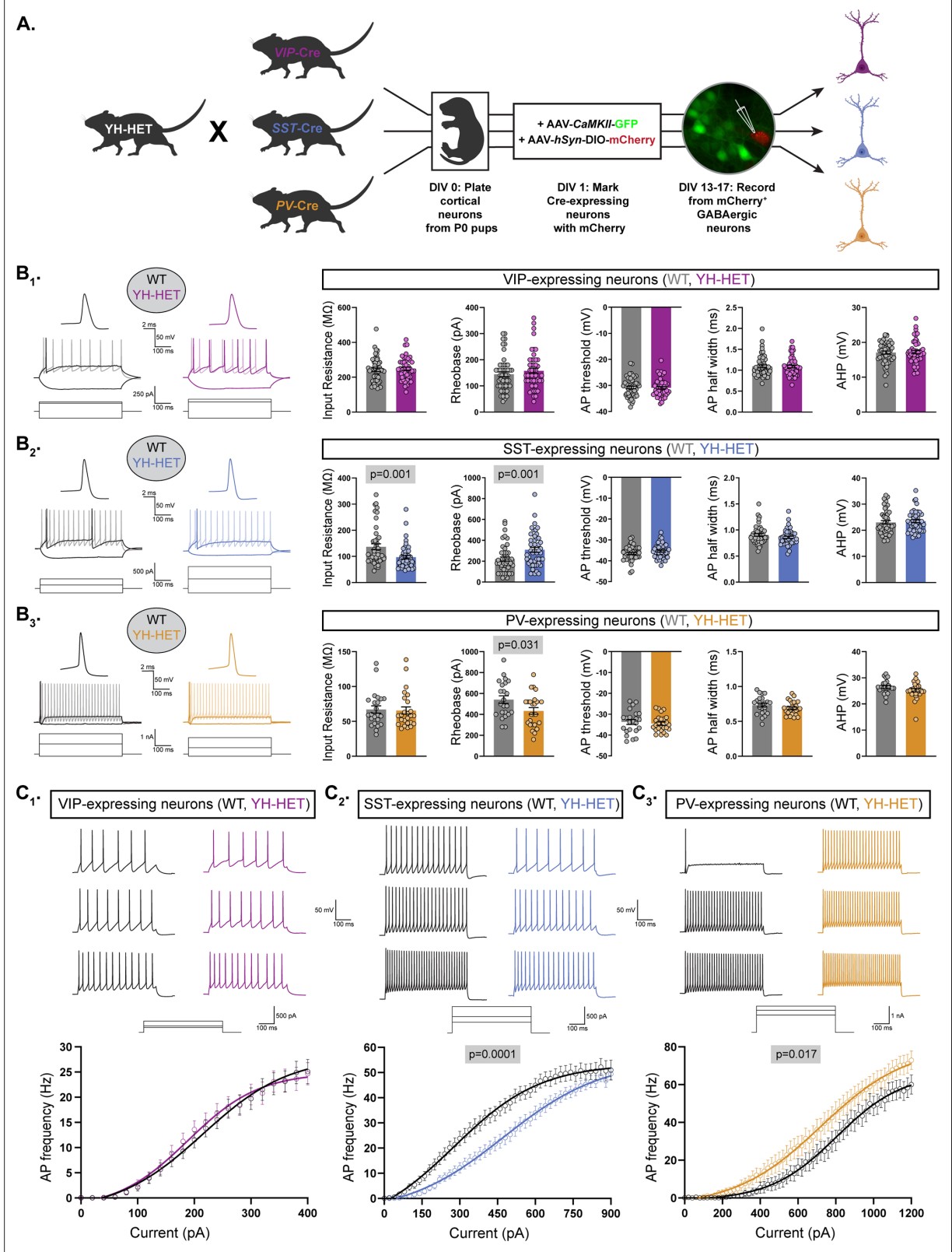

**Figure 4.** Heterozygous *Kcnt1*[Y777H] expression differentially affects the intrinsic excitability of somatostatin (SST)- and parvalbumin (PV)-expressing GABAergic neurons. (**A**) A schematic diagram illustrates the strategy for generating fluorescently labeled GABAergic subtype-specific neurons. YH-HET mice were crossed to vasoactive intestinal polypeptide (VIP)-, SST-, or PV-Cre mice, and the resulting P0 wildtype (WT) and YH-HET littermate pups were used to isolate and culture cortical neurons. At DIV 1, neurons were infected with AAV-*CamKIIa*-GFP to label glutamatergic neurons, and AAV-*hSyn*-

*Figure 4 continued on next page*

*Figure 4 continued*

DIO-mCherry to label Cre-expressing neurons. At DIV 13–17, whole-cell, patch-clamp electrophysiology was performed on mCherry$^+$/GFP$^-$ neurons. (B$_1$–B$_3$) On the left, representative responses to step currents are shown for VIP-, SST-, and PV-expressing WT (black) and YH-HET (colors) neurons (top to bottom). For each neuron type, the superimposed dark traces illustrate the input resistance (in response to a hyperpolarizing step) and the rheobase (the first trace with an action potential (AP) in response to a depolarizing step), and the light trace shows the first step current response to induce repetitive AP firing across the step. Above the superimposed traces, the first AP of each rheobase trace is shown (same vertical scale, increased horizontal scale). On the right, bar graphs show quantification of the membrane properties and AP parameters for each neuron type (VIP, SST, and PV, top to bottom) for WT (gray) and YH-HET (colors) groups, with individual neuron measurements overlaid in scatter plots. The p-values are shown above each graph where p < 0.05. (C$_1$–C$_3$) For VIP-, SST-, and PV-expressing neurons (left to right), representative traces are shown at low, medium, and high current steps, and the line graphs below show the number of APs (mean ± SEM) per current injection step in WT (black) and YH-HET (colors) neurons. Statistical significance was tested using generalized linear mixed models, and p-values are shown above each graph where p < 0.05. Data from this figure are from the following mouse pup (N) and neuron (n) sample sizes: VIP WT (N=3, n=50) and YH-HET (N=4, n=48), SST WT (N=4, n=45) and YH-HET (N=5, n=46), and PV WT (N=3, n=23) and YH-HET (N=4, n=25).

The online version of this article includes the following source data and figure supplement(s) for figure 4:

**Source data 1.** File containing data used to generate the graphs in *Figure 4*.

**Figure supplement 1.** Cre recombinase-expressing, cortical GABAergic neuron subpopulations retain characteristic electrophysiological features in culture.

**Figure supplement 1—source data 1.** File containing data used to generate the graphs in *Figure 4—figure supplement 1*.

**Figure supplement 2.** YH-HET somatostatin (SST) neurons have increased soma size compared with those of wildtype (WT).

**Figure supplement 2—source data 1.** File containing data used to generate the graphs in *Figure 4—figure supplement 2*.

**Figure supplement 3.** YH-HET and wildtype (WT) parvalbumin (PV) neurons show similar neurophysiological responses to increased temperature.

**Figure supplement 3—source data 1.** File containing data used to generate the graphs in *Figure 4—figure supplement 3*.

and PV neurons (*Figure 4—figure supplement 1B, C*, blue). Moreover, as reported previously, SST neurons showed the most hyperpolarized AP threshold of the three groups. Together, these data demonstrate the ability of cortical GABAergic neuron subtypes to retain their characteristic ex vivo passive and active membrane properties in vitro.

Next, we assessed the effects of the *Kcnt1*$^{Y777H}$ variant on the passive and active membrane properties of each GABAergic subtype. Current-clamp recordings from YH-HET VIP neurons revealed no significant effect of the variant on any of the membrane or AP properties measured (*Figure 4B$_1$*). Conversely, YH-HET SST neurons showed a strong decrease in input resistance, with an accompanying increase in rheobase current, relative to those of WT SST neurons (*Figure 4B$_2$*), similar to the hypoexcitable membrane phenotype observed in the YH-HET NFS neurons, without altering AP shape parameters. YH-HET SST neurons also showed a reduction in the membrane time constant and an increase in membrane capacitance, which was accompanied by an increase in soma size (*Supplementary file 2* and *Figure 4—figure supplement 2*), compared with those of WT SST neurons. Unexpectedly, YH-HET PV neurons exhibited a decrease in the rheobase current compared with that of WT (*Figure 4B$_3$*), and an increase in AP amplitude (*Supplementary file 2*), suggesting the *Kcnt1*$^{Y777H}$ variant increases PV neuron excitability.

Next, we measured AP firing frequency with incremental, 500 ms current steps in each of the three GABAergic subtypes. WT neurons showed AP firing frequencies similar to those reported previously from ex vivo recordings of neurons from the three Cre lines (*Taniguchi et al., 2011*), with VIP showing the lowest maximal firing frequency, PV the highest, and SST intermediate between VIP and PV (*Figure 4C$_{1-3}$* and *Supplementary file 2*). Consistent with a lack of any effects on membrane and AP shape properties, the AP firing frequencies across increasing current steps were indistinguishable between WT and YH-HET VIP neurons (*Figure 4C$_1$*). As expected, based on their hypoexcitable membrane properties, the YH-HET SST neurons fired fewer APs across all current steps relative to WT SST neurons (*Figure 4C$_2$*). In contrast to SST neurons, but consistent with their decreased rheobase current, YH-HET PV neurons fired more APs across all current steps relative to their WT counterparts (*Figure 4C$_3$*). Notably, although these experiments were performed at room temperature (~22°C), and increasing temperatures to 35°C altered multiple neuronal physiology parameters (*Figure 4—figure supplement 3A*), both WT and YH-HET PV neurons responded similarly to increased temperature (*Figure 4—figure supplement 3B*), suggesting that the observed phenotypes are not temperature dependent. Together, these data showed that, as hypothesized, KCNT1 GOF indeed strongly reduces

the excitability of SST neurons, but unexpectedly, it also causes a hyperexcitable effect in PV neurons; thus, the same ion channel mutation can lead to opposite effects on excitability in the two largest GABAergic neuron subtypes.

## The *Kcnt1*^Y777H variant increases KCNT1-mediated currents across subthreshold voltages in SST- and PV-expressing GABAergic neurons

Are the observed differential effects of KCNT1 GOF on GABAergic neuronal physiology due to distinct patterns of $K_{Na}$ current increases among GABAergic neuron subtypes? To answer this question, we first recorded $K_{Na}$ currents from GABAergic neurons cultured and labeled as described above, by applying voltage steps to voltage-clamped neurons and comparing the delayed outward current before and after the addition of TTX (*Figure 5—figure supplement 1A_{1–3}*). In all three subtypes, for both WT and YH-HET neurons, we observed $K_{Na}$ currents beginning around −10 mV and increasing with depolarization (*Figure 5—figure supplement 1B_{1–3}*). For YH-HET VIP neurons, pairwise comparisons to assess voltage-dependent differences showed that their $K_{Na}$ currents were not different from those of WT at any voltage step (*Figure 5—figure supplement 1B_1, C_1*). On the other hand, YH-HET SST neurons showed significant increases in $K_{Na}$ currents compared with those of WT across multiple subthreshold voltage steps, including −60, −50, and −40 mV (*Figure 5—figure supplement 1B_2, C_2*), similar to those observed in NFS GABAergic neurons with both heterozygous (*Figure 2*) and homozygous expression of the *Kcnt1*^Y777H variant (*Shore et al., 2020*), indicating a direct current-to-phenotype relationship in YH-HET SST neurons. Somewhat unexpectedly, pairwise comparisons showed that $K_{Na}$ currents were not different between YH-HET and WT PV neurons at any voltage step (*Figure 5—figure supplement 1B_3, C_3*).

The lack of an increase in TTX-sensitive currents in PV neurons, despite alterations in rheobase, AP amplitude, and AP firing frequency, suggests that either a TTX-insensitive $Na^+$ source activates KCNT1, or that compensatory alterations in an opposing current mask an increase in $K_{Na}$ and lead to the changes in AP firing. In past studies, the lack of selective KCNT1-specific inhibitors necessitated the use of indirect methods, such as TTX application or $Na^+$ replacement, to estimate the magnitude of $K_{Na}$. However, more recently, we identified and validated a small-molecule, selective KCNT1 channel inhibitor termed VU0606170 or VU170 (*Spitznagel et al., 2020*). To further validate the specificity of this small molecule, we applied voltage steps to voltage-clamped neurons lacking KCNT1 and KCNT2 channels, isolated and cultured from *Kcnt1*^−/−; *Kcnt2*^−/− double knockout mice (*Martinez-Espinosa et al., 2015*), and we compared the delayed outward current before and after the addition of either 10 µM VU170 or 0.5 µM TTX. This analysis indicated a non-specific outward VU-sensitive current that started around −10 mV and peaked at +50 mV (~1 nA peak current compared with ~3 nA in WT neurons), which was not present using TTX subtraction (*Figure 5—figure supplement 2A*). Across negative potentials, there was an inward TTX-sensitive current, which again is likely due to the persistent $Na^+$ current, whereas there was little to no outward VU-sensitive current (*Figure 5—figure supplement 2B*), validating the utility of VU170 in isolating $K_{Na}$ currents across this voltage range.

To better assess KCNT1-mediated currents in GABAergic subpopulations and disentangle potential confounds of using TTX, particularly across negative potentials, we applied the same voltage step protocol as in the TTX subtraction experiments but applied 10 µM VU170 instead. In all three WT GABAergic neuron subtypes, subtraction of the VU170 trace from the control trace revealed nA-sized outward currents at depolarized potentials, and in YH-HET neurons, the overall VU170-sensitive current was increased relative to those of WT, as measured by a significant effect of genotype using a linear model (*Figure 5A_{1–3}, B_{1–3}*). Pairwise comparisons at each voltage step showed voltage-dependent differences in current increases among the YH-HET neuron subtypes. For YH-HET VIP neurons, a significant increase in the VU170 current was only observed at +50 mV (*Figure 5B_1*), whereas subthreshold currents were indistinguishable from those of WT (*Figure 5C_1*). Previously, we similarly observed $K_{Na}$ current increases only at more positive potentials in YH-HOM glutamatergic neurons, and like YH-HET VIP neurons, their membrane and AP properties were unaltered by expression of the YH variant (*Shore et al., 2020*). Conversely, in YH-HET SST neurons, significant increases occurred at more negative voltage steps, from −60 to −30 mV (*Figure 5C_2*). These changes were similar to those observed with TTX treatment, but slightly larger, possibly due to the lack of the counteracting effect of the persistent $Na^+$ current, which is also blocked by TTX. In contrast to the TTX results, YH-HET PV neurons showed an increase in VU170-sensitive currents, with significant increases

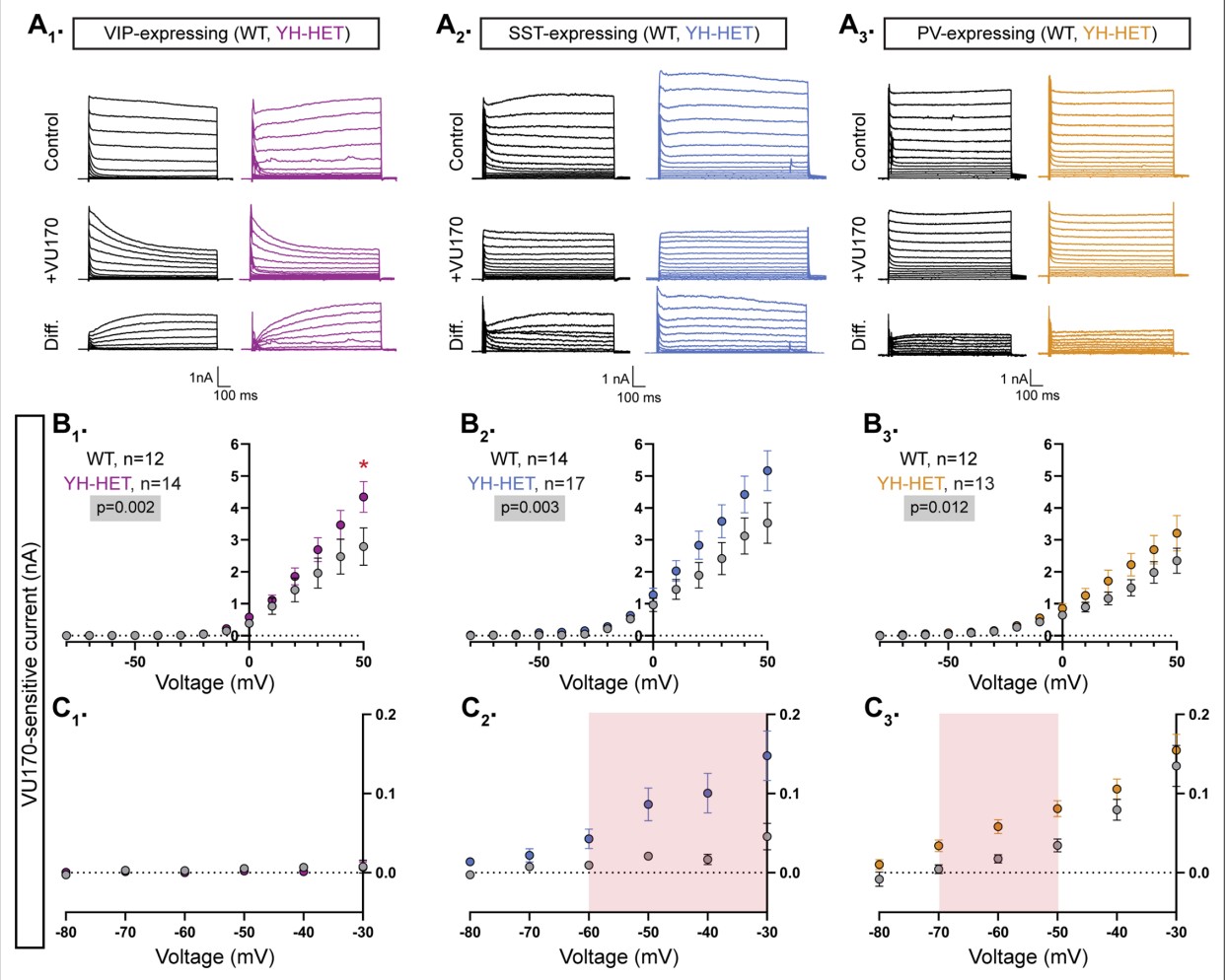

**Figure 5.** The *Kcnt1*[Y777H] variant increases KCNT1-mediated currents across subthreshold voltages in somatostatin (SST)- and parvalbumin (PV)-expressing GABAergic neurons. (**A₁–A₃**) Representative traces in control (top), 10 μM VU170 (middle), and the difference current (bottom) calculated by subtracting the membrane current response to voltage steps (−80 to +50 mV) from a holding potential of −70 mV in VU170 from the response in control external solution in vasoactive intestinal polypeptide (VIP)-, SST-, and PV-expressing, wildtype (WT) (black) and YH-HET (colors) neurons. To include all of the representative traces in the figures, and prevent overlap of the traces, the large inward sodium currents were removed from each set of traces using the masking tool in Adobe Illustrator. (**B₁–B₃**) Summary data show the KCNT1 current (mean ± SEM) for each voltage step in VIP-, SST-, and PV-expressing, WT (black and gray) and YH-HET (colors) neurons. The p-values are shown on each graph where p < 0.05, and the *n* values are the number of neurons recorded for each group. (**C₁–C₃**) Plots of the $K_{Na}$ current (mean ± SEM) for each voltage step from −80 to 0 mV in WT (black and gray) and YH-HET (colors) neurons to illustrate the values that are too small to be seen on the graphs in B₁–B₃. The shaded red areas in C₂ and C₃ indicate the subthreshold voltage range with significantly higher $K_{Na}$ currents (p < 0.05) in YH-HET relative to WT neurons. Statistical significance for *I–V* plots was tested using generalized linear mixed models with genotype and voltage step as fixed effects followed by pairwise comparisons at each level. Asterisks indicate where p < 0.05 (*).

The online version of this article includes the following source data and figure supplement(s) for figure 5:

**Source data 1.** File containing data used to generate the graphs in *Figure 5*.

**Figure supplement 1.** The *Kcnt1*[Y777H] variant increases $K_{Na}$ currents measured with tetrodotoxin (TTX) subtraction across subthreshold voltages in somatostatin (SST)-expressing GABAergic neurons.

**Figure supplement 1—source data 1.** File containing data used to generate the graphs in *Figure 5—figure supplement 1*.

**Figure supplement 2.** Isolation of $K_{Na}$ currents from neurons lacking KCNT1 and KCNT2 channels illustrates the specificity of VU170- vs. tetrodotoxin (TTX)-subtraction methods.

**Figure supplement 2—source data 1.** File containing data used to generate the graphs in *Figure 5—figure supplement 2*.

from −70 to −50 mV (*Figure 5C₃*). Taken together, these data identify distinct patterns of $K_{Na}$ current increases among GABAergic subtypes, and for VIP and SST neurons, these current increases are consistent with the observed effects of KCNT1 GOF on neuronal physiology. On the other hand, PV neurons showed subthreshold-specific $K_{Na}$ current increases that were highly similar to, and overlapping with, those of SST neurons; thus, differential $K_{Na}$ current increases alone likely do not account for the opposite effects of KCNT1 GOF on physiology observed in these two neuron types.

## Compartmental models of KCNT1 GOF in SST, but not PV, neurons are consistent with experimental data

The VU170-subtraction experiments showed that YH-HET SST and PV neurons have similar subthreshold increases in KCNT1-mediated currents, even though they exhibited opposing effects on neuron excitability and AP generation, and that YH-HET VIP neurons have suprathreshold increases in KCNT1-mediated currents with no effects on neuron physiology. Because GABAergic subtypes exhibit varying morphologies and express unique repertoires of ion channels, which are known to give rise to their characteristic membrane and AP firing properties, we hypothesized that at least some of the observed differential effects of KCNT1 GOF, in particular the opposite effects in SST and PV neurons, are due to these inherent, neuron-type-dependent differences. To test this hypothesis, we simulated the effect of KCNT1 GOF in compartmental models of these three cortical neuron types. We used a KCNT1 conductance with the $Na^+$ dependence constrained by prior studies in neurons ($EC_{50}$ = 40 mM, slope 3.5; see Materials and methods) and the voltage dependence based on the activation curves from our own experimental data of the VU170-sensitive current in each subtype (*Figure 6A*). The kinetics of the model current were set by measuring the onset time course of the VU170-sensitive current in outside-out membrane patches of each neuron type (*Figure 6B, C*). This conductance was inserted into compartmental models of 10 neurons per subtype, representing a variety of morphological and electrical properties of each subtype (*Markram et al., 2015*; https://bbp.epfl.ch/nmc-portal/downloads.html). Because the Y796H variant was previously shown to increase the $Na^+$ sensitivity of the channel, we modeled the GOF effect by reducing the $EC_{50}$ for $Na^+$ activation to two levels (35 and 30 mM), which resulted in moderate $K_{Na}$ current increases above WT levels (*Figure 6D*).

Consistent with our experimental data, introducing KCNT1 GOF at the lower level ($EC_{50}$ = 35 mM) into model VIP neurons did not alter their AP firing rate, input resistance, or rheobase, whereas at the higher level ($EC_{50}$ = 30 mM), there were small changes in input resistance (40 mM: 266.5 ± 28.2; 30 mM: 264.8 ± 28.0, p = 0.012) and rheobase (40 mM: 78.7 ± 11.6; 30 mM: 80.4 ± 11.1, p = 0.014), but these changes were not sufficient to affect the AP firing rate (*Figure 6E₁, E₂*, *Supplementary file 3*, and *Figure 6—figure supplement 1A*). On the other hand, model SST neurons with KCNT1 GOF at both levels fired fewer APs per current step as the GOF was increased, and showed significantly reduced input resistances, higher rheobases, and lower AP firing rates (*Figure 6F₁, F₂* and *Supplementary file 3*), results that agree with our experimental observations. In contrast to experimental data, but similar to model SST neurons, model PV neurons responded to KCNT1 GOF with decreased input resistance, increased rheobase, and a reduced AP firing rate (*Figure 6G₁, G₂* and *Supplementary file 3*), although the magnitude of these effects were smaller than those in model SST neurons (*Figure 6—figure supplement 1A*). These data suggest that the increased excitability observed in YH-HET PV neurons is not simply due to an intrinsic property of this GABAergic subtype, but instead results from an indirect mechanism (e.g. a compensatory response like an increase in $Na^+$ channels) or depends on a feature of KCNT1 function not captured by the model.

Lastly, we hypothesized that the lack of GOF effects on VIP and glutamatergic neuron physiology is due to the altered kinetics of their KCNT1 channel activation curves, in particular their right shifted $V_{50}$ and decreased slope, relative to those of SST or PV neurons (*Figure 6A*). To test this hypothesis, we performed simulations in which the activation curve parameters ($V_{50}$ and slope) measured in SST neurons were inserted into VIP and glutamatergic neurons, with the $Na^+$ $EC_{50}$ set to 30 mM to simulate KCNT1 GOF (*Figure 6—figure supplement 2A₁, B₁*, blue traces). Inserting KCNT1 channels with SST kinetics into VIP and glutamatergic neurons indeed enhanced the GOF effects on input resistance and rheobase, and importantly, resulted in a significantly decreased AP firing rate relative to control neurons for both neuron types (*Figure 6—figure supplement 2A₂, B₂*, *Supplementary file 3*, and *Figure 6—figure supplement 1B*). This model-based evidence suggests that the relative sensitivity of

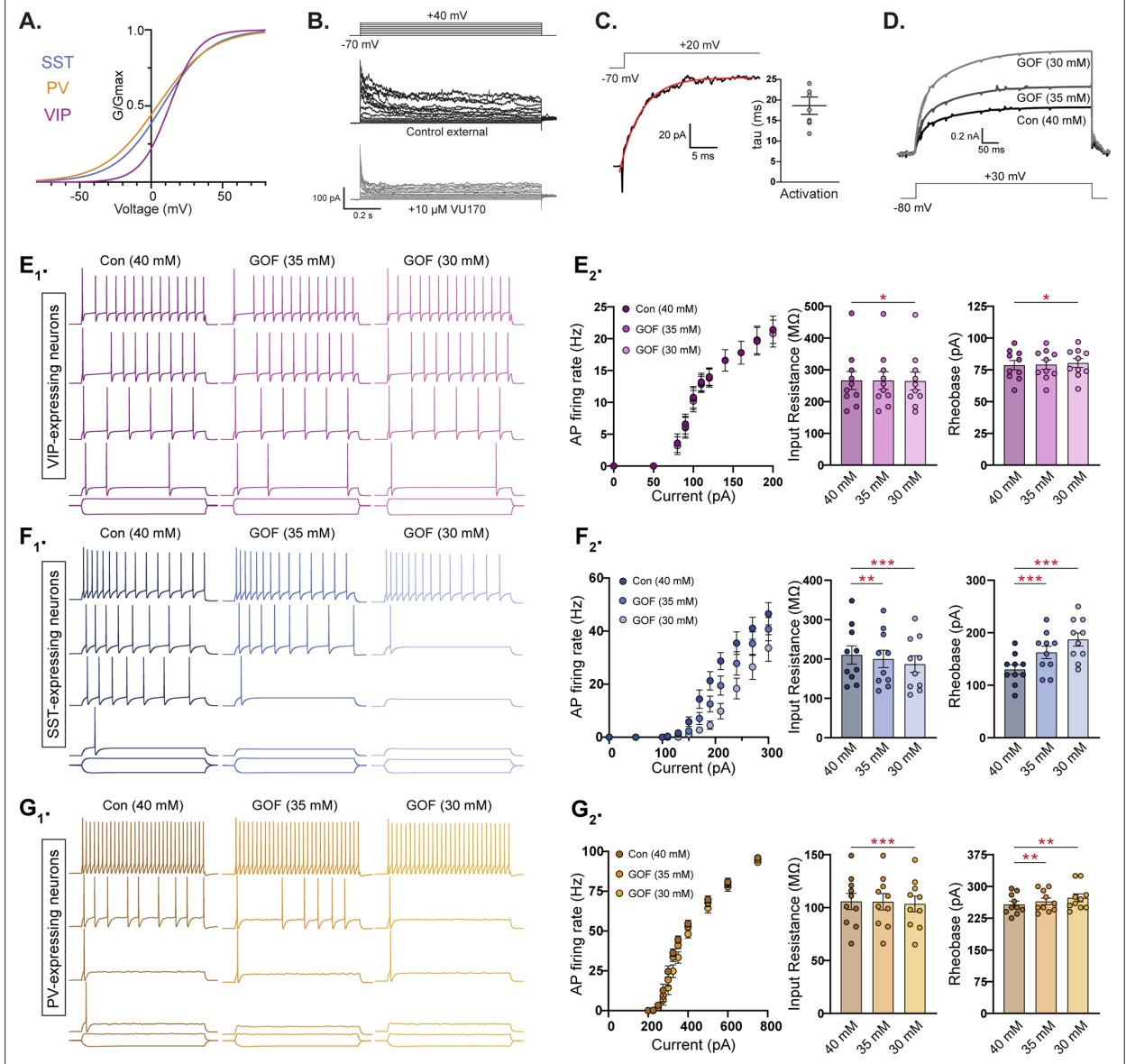

**Figure 6.** Compartmental models of KCNT1 gain-of-function (GOF) in somatostatin (SST), but not parvalbumin (PV), neurons are consistent with experimental data. (**A**) Activation curves for vasoactive intestinal polypeptide (VIP), SST, and PV neurons used to model the KCNT1 channel in each neuron type. (**B**) Representative traces of the response of an outside-out membrane patch pulled from an SST neuron in control external solution (top) and after application of 10 µm VU170 (bottom). (**C**) Representative trace of the VU170-sensitive current in an outside-out patch pulled from an SST neuron with a single exponential fit (red curve) overlaid and summary data showing the time constant (mean ± SEM) obtained from these fits. (**D**) Simulated traces showing the KCNT1-mediated current in model neurons at three levels of $Na^+$-sensitivity (Con-40 mM, GOF-35 mM, and GOF-30 mM). (**E₁, F₁, and G₁**) Simulated traces from model VIP (**E₁**; purple), SST (**F₁**; blue), and PV (**G₁**; yellow) neurons in response to 500 ms current injections at three levels of $Na^+$ sensitivity [control (Con), and two levels of KCNT1 GOF]. Representative traces are shown at increasing current steps from bottom to top for each level. (**E₂, F₂, and G₂**) Summary data from 10 model VIP (**E₂**; purple), SST (**F₂**; blue), and PV (**G₂**; yellow) neurons showing (from left to right) the number of action potentials (APs) at different current steps (F–I plot), input resistance, and rheobase. The bar graphs (mean ± SEM) are overlaid with scatter plots of individual neuron measurements. Statistical significance was tested using a repeated measures analysis of variance (ANOVA). Asterisks indicate where $p < 0.05$ (*), $p \leq 0.01$ (**), and $p \leq 0.001$ (***).

The online version of this article includes the following source data and figure supplement(s) for figure 6:

**Source data 1.** File containing data used to generate the graphs in *Figure 6*.

**Figure supplement 1.** Model neurons show differential responses to KCNT1 gain-of-function (GOF).

**Figure supplement 1—source data 1.** File containing data used to generate the graphs in *Figure 6—figure supplement 1*.

*Figure 6 continued on next page*

Figure 6 continued

**Figure supplement 2.** Model vasoactive intestinal polypeptide (VIP) and glutamatergic neurons with insertion of somatostatin (SST)-like, KCNT1 gain-of-function (GOF) kinetics show reduced excitability.

**Figure supplement 2—source data 1.** File containing data used to generate the graphs in *Figure 6—figure supplement 2*.

SST and PV neurons, and resistance of VIP and glutamatergic neurons, to KCNT1 GOF is due to the differences in voltage dependence of the $K_{Na}$ current in each subtype.

## The persistent Na⁺ current is increased by the *Kcnt1*^Y777H variant in PV, but not SST, neurons

The modeling results suggest that a subthreshold increase in the $K_{Na}$ current alone is not sufficient to account for the alterations in PV neuron excitability. Moreover, the finding of an increase in the KCNT1-mediated current in YH-HET PV neurons when measured with VU170, but not TTX, isolation suggests there may be a compensatory upregulation of an opposing current that masks the increase in $K_{Na}$ and leads to the unexplained increase in AP firing. The persistent Na⁺ current ($I_{NaP}$) is active on the same time scale as $K_{Na}$ and was previously shown to provide the Na⁺ source for $K_{Na}$ (***Hage and Salkoff, 2012***); therefore, we hypothesized that KCNT1 GOF causes an increase in $I_{NaP}$ in PV neurons that enhances their excitability. To test this hypothesis, we measured $I_{NaP}$ in SST and PV neurons by applying TTX during slow voltage ramp protocols (20 mV/s) designed to isolate $I_{NaP}$ from both the transient Na⁺ current and the $K_{Na}$ current (***Figure 7—figure supplement 1***). In SST neurons, $I_{NaP}$ did not differ between WT and YH-HET neurons, either in the shape of the response (***Figure 7A, B***), or the peak amplitude (***Figure 7B***). In PV neurons, however, the peak amplitude of $I_{NaP}$ was significantly increased by the *Kcnt1*^Y777H variant (***Figure 7C, D***). Interestingly, the mean peak amplitude of $I_{NaP}$ in WT PV neurons was 70% larger than that in WT SST neurons (−1.42 ± 0.16 to −0.85 ± 0.07 pA/pF; ***Figure 7B, D***), suggesting there may be differences in sodium channel expression, localization, or regulation inherent to each neuron type that confer their differential response to KCNT1 GOF.

To examine whether the observed increase in $I_{NaP}$ in YH-HET PV neurons could account for the increase in AP firing, we again simulated the effect of KCNT1 GOF (Na⁺ $EC_{50}$ = 35 mM) on PV neuron activity, but this time included an increase in $I_{NaP}$ in the compartmental models. Indeed, an increase in $I_{NaP}$ conductance, similar to what was seen in the experimental results (twofold), was sufficient to increase the number of APs fired in response to increasing current steps (***Figure 7E, F*** and ***Supplementary file 3***), even in the face of the higher level of KCNT1 GOF (30 mM; ***Supplementary file 3*** and ***Figure 7—figure supplement 2***). Also, like the experimental data, the rheobase current was decreased (***Figure 7F*** and ***Figure 7—figure supplement 2***) and the AP height was increased (80.4 ± 0.7 vs. 82.1 ± 0.8 mV). These results suggest that the differential effects on AP firing in SST and PV neurons can be accounted for by the absence or presence of a secondary increase in $I_{NaP}$.

## YH-HET SST neurons show increased chemical and electrical coupling and receive increased excitatory input

We previously showed that homozygous expression of the *Kcnt1*^Y777H variant increased connections between GABAergic neurons (***Shore et al., 2020***), but heterozygous expression did not (***Figure 3***). Because heterozygous *Kcnt1*^Y777H variant expression caused the strongest excitability impairments in SST neurons, we hypothesized that alterations in synaptic connectivity and activity may be more likely to occur in this neuronal population. To test this hypothesis, we recorded from pairs of WT or YH-HET SST neurons as described above to measure connection probability and strength. Indeed, similar to the increase observed in I–I connections in YH-HOM networks, we found an increase in homotypic synaptic connections of SST neurons in YH-HET networks (30/40 connections) relative to those in WT (15/40 connections; ***Figure 8A***). The amplitudes of the evoked postsynaptic currents (ePSCs) between connected SST neurons were not significantly different between the YH-HET and WT groups (***Figure 8B***).

There are two main types of synaptic coupling between neurons: chemical, which is mediated by neurotransmitter release, and electrical, which is mediated by gap junctions. To test for alterations in each type of synaptic coupling due to expression of the *Kcnt1* variant, we recorded from 20 pairs of WT or YH-HET SST neurons in close proximity (<100 μm apart). Under current-clamp

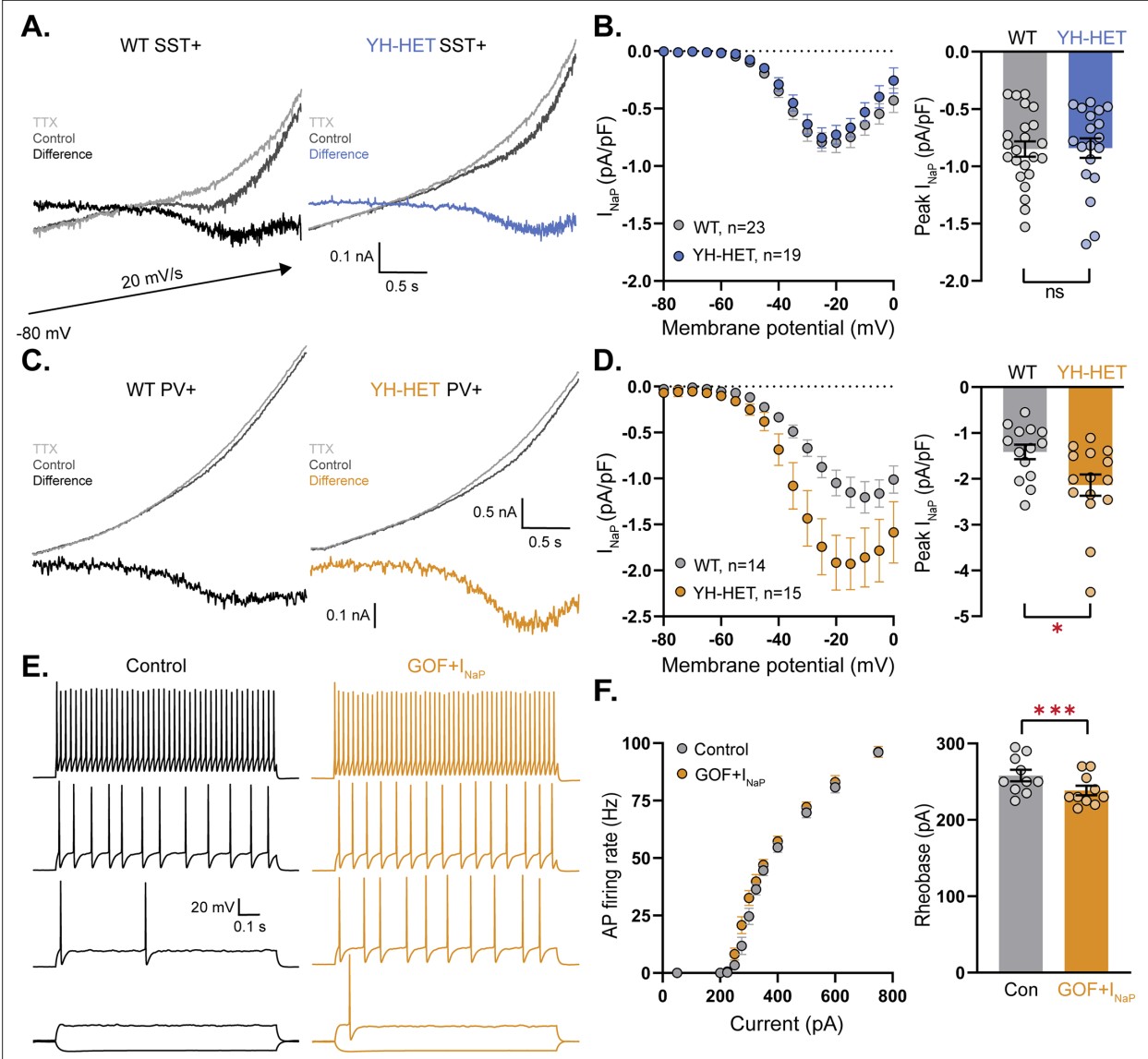

**Figure 7.** The persistent Na⁺ current is increased by the *Kcnt1*^Y777H variant in parvalbumin (PV), but not somatostatin (SST), neurons. (**A**) Representative traces of wildtype (WT) (left) and YH-HET (right) SST neurons in response to slow voltage ramp in control (dark gray) and 7 s after application of 0.5 nM tetrodotoxin (TTX) (light gray) with the difference current ($I_{NaP}$) plotted on the same scale. (**B**) $I_{NaP}$ *I–V* curves of WT (gray) and YH-HET (blue) SST neurons constructed from averaging the ramp-evoked difference current (mean ± SEM) at 5 mV intervals (left), and a bar graph with overlaid scatter plot showing the peak negative value for each neuron (right). (**C**) Representative traces of WT (left) and YH-HET (right) PV neurons in response to slow voltage ramp in control (dark gray) and 7 s after application of 0.5 nM TTX (light gray) with the difference current ($I_{NaP}$) plotted on the same scale. The upper traces and difference currents are plotted on different scales. (**D**) $I_{NaP}$ *I–V* curves of WT (gray) and YH-HET (orange) PV neurons constructed from averaging the ramp-evoked difference current (mean ± SEM) at 5 mV intervals (left), and a bar graph with overlaid scatter plot showing the peak negative value for each neuron (right). (**E**) Simulated traces from a model PV neuron in response to 500 ms current injections with control levels of KCNT1 (black traces) and KCNT1 gain-of-function (GOF) (Na⁺ EC$_{50}$ = 35 mM) with a twofold increase in $I_{NaP}$ (orange traces). Representative traces are shown at increasing current steps from bottom to top for each level. (**F**) *F–I* plot shows the increase in action potential (AP) firing (left), and a bar graph with overlaid scatter plot of individual neuron values shows the decrease in rheobase associated with modeling KCNT1 GOF (Na⁺ EC$_{50}$ = 35 mM) together with the increase in $I_{NaP}$. For modeling data, statistical significance was tested using a repeated measures ANOVA. Asterisks indicate where $p < 0.05$ (*) and $p \leq 0.001$ (***).

The online version of this article includes the following source data and figure supplement(s) for figure 7:

**Source data 1.** File containing data used to generate the graphs in *Figure 7*.

**Figure supplement 1.** The rate of K$_{Na}$ current block after tetrodotoxin (TTX) application is slow relative to $I_{NaP}$ block in GABAergic neurons.

**Figure supplement 1—source data 1.** File containing data used to generate the graphs in *Figure 7—figure supplement 1*.

**Figure supplement 2.** Model parvalbumin (PV) neurons show differential responses to KCNT1 gain-of-function (GOF) with increased $I_{NaP}$.

**Figure supplement 2—source data 1.** File containing data used to generate the graphs in *Figure 7—figure supplement 2*.

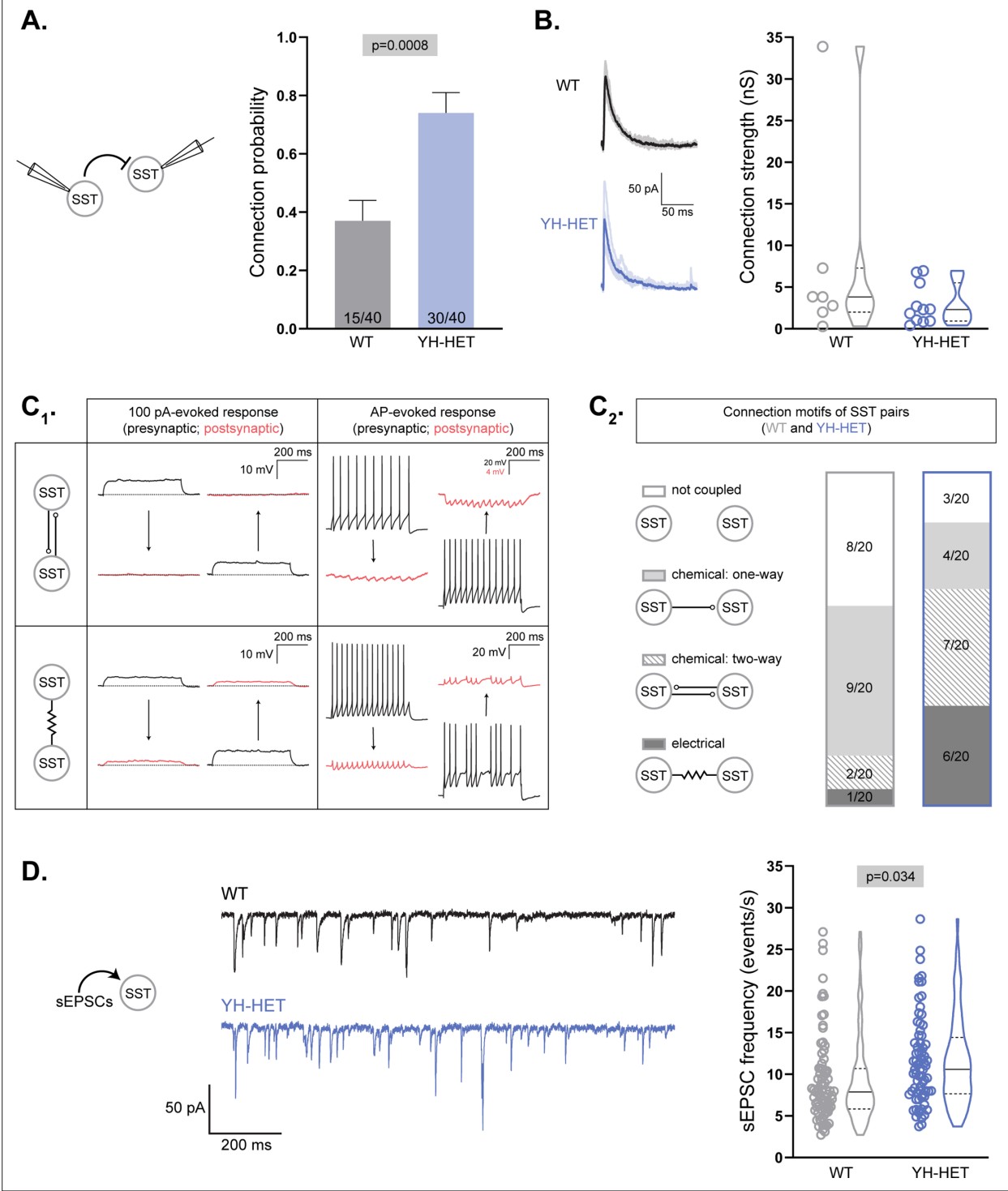

**Figure 8.** YH-HET somatostatin (SST) neurons show increased chemical and electrical coupling and receive increased excitatory input. (**A, B**) Evoked postsynaptic currents (PSCs) were recorded from SST neuron pairs by stimulating each neuron at 0.1 Hz and recording the response onto the partner. (**A**) Bar graph shows the connection probability (mean + SEM) between SST neurons (wildtype [WT], gray; YH-HET, blue; numbers on bars represent connected pair number/recorded pair number). (**B**) On the left, example traces of evoked IPSCs [individual IPSCs (light) overlaid by averaged IPSCs (dark)], and on the right, a graph of connection strength between SST neurons (WT, gray; YH-HET, blue; individual measurements and summary violin plots). (**C₁**) A schematic illustrates representative responses (presynaptic, black; postsynaptic, red) between chemically coupled (upper two panels) and electrically coupled (lower two panels) SST neurons following 100 pA (left two panels) and action potential (AP) (right two panels) stimulation. (**C₂**) Summary data of four possible connection motifs (not coupled, chemical: one-way, chemical: two-way, and electrical) tested among WT (gray box) and

*Figure 8 continued on next page*

*Figure 8 continued*

YH-HET (blue box) SST neuron pairs (20 pairs/group). (**D**) On the left, example traces of spontaneous EPSCs (sEPSCs) recorded onto SST neurons (WT, black; YH-HET, blue). On the right, a graph shows individual neuron measurements and summary violin plots of the sEPSC frequency onto SST neurons (WT, gray; YH-HET, blue). All significant p-values are displayed at the top of each graph.

The online version of this article includes the following source data for figure 8:

**Source data 1.** File containing data used to generate the graphs in *Figure 8*.

conditions, we injected a series of hyperpolarizing and depolarizing current steps into one neuron and assessed whether the current induced a simultaneous voltage deflection in the paired, non-injected neuron (*Figure 8C₁*; 100-pA-evoked response). We then evoked a train of APs in one neuron and assessed the AP-induced voltage changes in the paired, non-injected neuron (*Figure 8C₁*; AP-evoked response). SST pairs were considered to be chemically coupled if the AP trains in one neuron induced corresponding IPSCs, without inducing voltage deflections in response to the current steps, onto the paired neuron (*Figure 8C₁*; top panels). SST pairs were considered to be electrically coupled if the current steps injected into one neuron induced simultaneous voltage deflections in the paired neuron (*Figure 8C₁*; bottom panel, left). Electrically coupled SST neurons also frequently showed AP-evoked spikelets onto paired neurons due to low-pass filtering (*Figure 8C₁*; bottom panel, right), which results in a greater attenuation of the high frequency portion (spike), than the low-frequency portion (AHP), of an AP as it passes through an electrical synapse.

Of the 20 WT SST neuron pairs tested, 12 were coupled, and the majority (9/12) of those were one-way chemical connections (*Figure 8C₂*). Of the 20 YH-HET SST neuron pairs tested, 17 were coupled, and the majority (13/17) of those were two-way, or bidirectionally, connected (*Figure 8C₂*). Of the 3 bidirectionally connected WT SST neuron pairs, 2 were chemical and 1 was electrical, whereas of the 13 bidirectionally connected YH-HET SST neuron pairs, 7 were chemical and 6 were electrical (*Figure 8C₂*). Thus, these data suggest that heterozygous $Kcnt1^{Y777H}$ expression increases reciprocal connectivity among SST neurons, of both the chemical and electrical sort.

Finally, although we previously found no alterations in the sEPSC frequency onto the total population of YH-HOM GABAergic neurons, there was an increase in the sEPSC frequency onto those with an NFS phenotype (*Shore et al., 2020*), suggesting there may be a compensatory increase in excitatory drive onto NFS YH-HOM GABAergic neurons to offset the effects of their decreased membrane excitability. To assess whether there is a similar increase in sEPSC frequency onto SST-expressing YH-HET neurons, which showed a similar decrease in membrane excitability to that of NFS YH-HOM GABAergic neurons, we recorded sEPSCs onto voltage-clamped WT and YH-HET SST neurons. Indeed, there was an increase in sEPSC frequency onto YH-HET SST neurons compared with that of WT (*Figure 8D*), suggesting a compensatory increase in excitatory drive onto the most impaired GABAergic subtype.

## Acute cortical slice recordings confirm the reduced intrinsic excitability of YH-HET SST neurons

Of the three GABAergic neuron populations assessed, SST-expressing neurons showed the strongest impairments, both direct and indirect, downstream of the heterozygous $Kcnt1^{Y777H}$ variant. Importantly, the observed effects on YH-HET SST neuron physiology were consistent with those that would be expected, based on YH-HET SST subthreshold $K_{Na}$ current increases (*Figure 5—figure supplement 1C₂*, and *Figure 5C₂*), and predicted, based on SST KCNT1 GOF neuron modeling data (*Figure 6F₁, F₂*). Thus, to test whether the most significant physiological alterations identified in cortical neuron cultures are also present in an experimental preparation with a more organized network, we crossed YH HET mice with homozygous GIN (GFP-expressing inhibitory neurons) mice, which express GFP in a subpopulation of SST-expressing neurons of the hippocampus and cortex (*Ma et al., 2006*; *Oliva et al., 2000*), and performed electrophysiological recordings in acute brain slices from WT and YH-HET progeny.

First, we assessed the effects of the heterozygous $Kcnt1^{Y777H}$ variant on the passive and active membrane properties by performing whole-cell, current-clamp recordings of layer 2/3, GFP-expressing neurons in motor and somatosensory cortical regions of slices prepared from P21 to P30 mice (*Figure 9A*). Incremental current injections to elicit APs demonstrated a strong impairment in

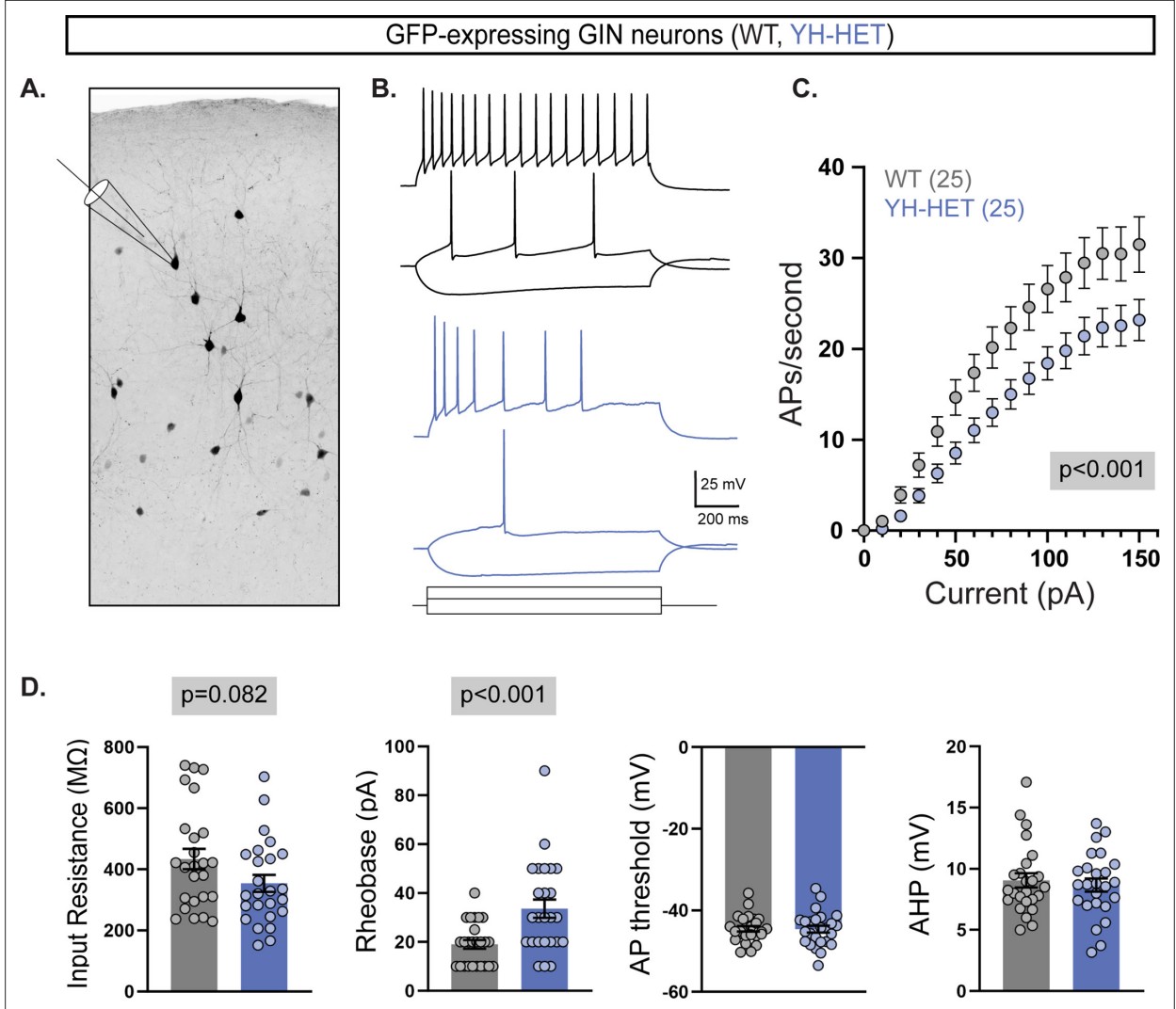

**Figure 9.** Acute cortical slice recordings confirm the reduced intrinsic excitability of YH-HET SST neurons. (**A**) An inverted monochrome image illustrates the recording scheme from GFP-expressing neurons in acute slices from P21 to P30 GIN mice that express GFP in a subset of SST-positive neurons. (**B**) Representative responses to current steps are shown for wildtype (WT) (black) and YH-HET (blue) to three, equivalent 1 s current steps. (**C**) Plot of the action potential (AP) frequency (mean ± SEM; 25 neurons/group) per current injection step shows that YH-HET GFP-expressing neurons (blue) fire fewer APs than those of WT (gray) across a range of current inputs. (**D**) Bar graphs show quantification of the membrane properties and AP parameters for WT (gray) and YH-HET (blue) GFP-expressing neuron groups, with individual neuron measurements overlaid in scatter plots. Relevant p-values are shown in gray boxes on or above graphs.

The online version of this article includes the following source data for figure 9:

**Source data 1.** File containing data used to generate the graphs in *Figure 9*.

AP generation downstream of the heterozygous *Kcnt1*^Y777H variant, as the YH-HET GFP-expressing neurons fired fewer APs across all current steps than the WT GFP-expressing neurons, similar to that observed in the YH-HET SST cultured neurons (*Figure 9B, C*). Although YH-HET GFP-expressing neurons in slice did not show a significant decrease in input resistance relative to those from WT, they did show a large increase in rheobase current, without alterations in AP shape (*Figure 9D*). Thus, we confirmed the main electrophysiological finding from culture—the reduced excitability of YH-HET SST neurons—in acute slices from YH-HET juvenile mice.

## Discussion

More than a decade ago, autosomal dominant mutations were identified in the sodium-gated potassium channel *KCNT1* in multiple patients with severe, childhood epilepsy syndromes (*Barcia et al., 2012*; *Heron et al., 2012*). *KCNT1*-related epilepsy patients suffer not only with frequent, early-onset seizures, but also with cognitive impairments, movement disorders, and sometimes additional behavioral and/or psychiatric problems (*Bonardi et al., 2021*). Unfortunately, after over 10 years of research efforts, *KCNT1*-related epilepsies remain highly refractory to current anti-seizure medications (*Bonardi et al., 2021*); thus, a better understanding of the pathological mechanisms underlying *KCNT1*-related epilepsies—from mutated gene to altered K⁺ current to altered neuronal physiology to altered neural network to seizure—is imperative for advancing therapeutic strategies to improve seizure control and enhance patient quality of life.

### Heterozygous vs. homozygous *KCNT1* GOF variant effects

Despite the heterozygous nature of *KCNT1* GOF variants in the overwhelming majority of *KCNT1*-related DEE patients, research efforts, including our own, have largely focused on homozygous GOF variant effects on channel function and neuronal physiology (*Gertler et al., 2019*; *Kim et al., 2014*; *McTague et al., 2018*; *Mikati et al., 2015*; *Milligan et al., 2014*; *Quraishi et al., 2019*; *Shore et al., 2020*; *Tang et al., 2016*). Heterozygous *KCNT1* GOF variant expression likely results in the formation of heteromeric KCNT1 channels, consisting of both WT and mutant subunits, whose characteristics and kinetics may differ from those of homomeric KCNT1 channels; thus, it is critical to distinguish between GOF variant effects in heteromeric and homomeric KCNT1 channels to identify the pathogenic mechanisms of *KCNT1* GOF variants in DEEs. Recent work in heterologous cells has shown that KCNT1 GOF heteromeric channels, with WT and mutant subunits, exhibit $K_{Na}$ current increases that lie between those of WT and KCNT1 GOF homomeric channels (*Cole et al., 2021*; *Dilena et al., 2018*; *Rizzo et al., 2016*). Furthermore, other recent studies have similarly shown that heterozygous *Kcnt1* GOF variant expression in neurons causes effects on $K_{Na}$ current and excitability that are intermediate to those of WT and homozygous variant-expressing neurons (*Gertler et al., 2022*; *Wu et al., 2024*).

Here, we assessed the effects of heterozygous expression of *Kcnt1*^Y777H^ on $K_{Na}$ currents, neuronal physiology, and synaptic connectivity among cortical neuron subtypes in mice, and compared the effects to those observed previously with homozygous expression of the same variant (*Shore et al., 2020*). We found that the heterozygous YH variant caused similar effects on $K_{Na}$ currents and neuronal physiology to those caused by the homozygous YH variant, including subthreshold-specific increases in $K_{Na}$ current, altered AP shape, and impaired AP firing of NFS GABAergic neurons, but with magnitudes intermediate to those of WT and YH-HOM neurons. At the network and whole-animal level, we previously showed that homozygous expression of the YH variant in mice increases homotypic synaptic connectivity (E–E and I–I) and the E/I ratio, resulting in hypersynchronous, hyperexcitable YH-HOM cortical networks and frequent seizures. Here, we found no evidence of similar network alterations in YH-HET cortical networks, which likely explains the infrequent seizures observed in the heterozygous YH mouse model, and together with the YH-HOM data, indicate a strong relationship between network abnormalities and seizure propensity downstream of the YH variant. Furthermore, these data suggest that heterozygous and homozygous *KCNT1* variant expression cause similar, gene-dose-dependent GOF effects at the current and neuron level, but not at the network or whole-animal level.

### Neuron-type-dependent *KCNT1* GOF variant effects

Prior work has illustrated the importance of cell context in delineating the functional effects of ion channel variants on neuronal physiology. For instance, several studies using LOF channelopathy models of both sodium and potassium channels have identified excitability effects on one neuron type without effects on another (*Hedrich et al., 2014*; *Soh et al., 2018*; *Tai et al., 2014*; *Yu et al., 2006*). Other studies have shown that the same ion channel variant can even cause opposite effects on excitability in different neuron types (*Makinson et al., 2016*; *Rush et al., 2006*; *Studtmann et al., 2022*). Thus, understanding how KCNT GOF alters networks to lead to hyperexcitability and seizures requires a thorough evaluation of how the variant impacts each neuron type that participates in the network. With this goal in mind, we assessed the effects of heterozygous *Kcnt1* GOF on $K_{Na}$ currents

and AP generation among cortical glutamatergic neurons and all three major GABAergic neuron subtypes, including those expressing VIP, SST, and PV.

Using two methods to isolate and measure the KCNT1-mediated current, VU170 and TTX subtraction, we observed voltage-dependent differences in current increases among the YH-HET neuron types. YH-HET glutamatergic and VIP neurons showed KCNT1-mediated current increases across suprathreshold voltages, whereas YH-HET SST and PV neurons had current increases across subthreshold voltages. These neuron-type-dependent effects of KCNT1 GOF on $K_{Na}$ current increases are similar to those observed previously in YH-HOM, glutamatergic and GABAergic neuron subpopulations using TTX (*Shore et al., 2020*). Moreover, these results are similar to those reported in a recent study using a different KCNT1 GOF mouse model (human R474H, mouse R455H), which showed $K_{Na}$ current increases selectively across positive potentials in glutamatergic, but across positive and negative potentials in FS and NFS GABAergic, neurons (*Wu et al., 2024*). The mechanism underlying these neuron-type-dependent KCNT1 GOF effects on $K_{Na}$ current increases is unknown, however, we have observed differences even in WT KCNT1 channel activation curves between glutamatergic and VIP neurons (steeper slopes of activation) and SST and PV neurons (shallower slopes of activation), suggesting there are factors affecting KCNT1 channel kinetics that are inherent to each neuron type (*Figure 6A*). Future studies should investigate these potential factors including (1) differential expression of alternative KCNT1 splice forms, some of which are known to have different activation kinetics, (2) differential expression of other channels such as KCNT2, which can form heteromers with KCNT1 and alter its biophysical properties, and (3) differential expression and/or localization of, and/or coupling to, sodium channels that act as the source of KCNT1 channel activation (*Chen et al., 2009*; *Hage and Salkoff, 2012*; *Joiner et al., 1998*).

Consistent with the differential KCNT1 GOF effects on $K_{Na}$ currents, we observed a range of effects on membrane properties and AP generation among the cortical neuron populations. We observed no effects on YH-HET glutamatergic and VIP neurons, whereas YH-HET SST neurons showed reduced AP firing, with decreased input resistance and increased rheobase, and YH-HET PV neurons showed increased AP firing, with decreased rheobase. As a complementary approach, we performed in silico electrophysiology using our measured neuron-type-specific KCNT1 channel activation kinetics, with and without GOF, for each neuron type. For glutamatergic and VIP neurons, the steeper slopes of activation limited the effects of GOF on intrinsic excitability, whereas for SST neurons, the shallower slope of activation decreased the input resistance, increased the rheobase, and impaired AP generation downstream of KCNT1 GOF. Each of these modeling results are similar to those observed using in vitro electrophysiology, indicating direct $K_{Na}$ current-to-KCNT1 GOF phenotype relationships in these neuron types.

Conversely, for PV neurons, unlike the experimental observation of increased YH-HET PV neuron excitability, in silico electrophysiology predicted a slight reduction in PV neuron excitability downstream of KCNT1 GOF. In support of this in silico prediction, in a different KCNT1 GOF mouse model, heterozygous expression of the human L456F variant indeed reduced AP firing of PV neurons, without altering any other membrane properties or AP parameters (*Gertler et al., 2022*). The opposite effect of KCNT1 GOF on PV excitability between these two studies may simply result from variant-specific alterations in KCNT1 channel kinetics; however, both GOF variants showed a similar lack of effect on membrane properties and AP generation of glutamatergic neurons. Alternatively, the opposite effects could be due to the different experimental conditions used in the two studies, including (1) different preps and mouse ages (acute slices from P42 to P60 mice vs. DIV 13 to 17 cultured neurons from P0 pups) and (2) different brain regions (hippocampal vs. cortical). The idea of brain region-dependent effects of ion channel variants is not unprecedented, as a recent study showed that a potassium channel GOF variant causes opposite effects on excitability in the same neuron type from different brain regions (hippocampal CA1 vs. cortical layer 2/3 pyramidal neurons), and this opposite effect was suggested to be due to the compensatory upregulation of sodium channels in one neuron type but not the other (*Varghese et al., 2023*). Taken together, these findings illustrate the importance of interrogating not only multiple neuron types but also multiple brain regions to fully unravel the complexity underlying KCNT1 GOF-related neurodevelopmental disorders.

## Evidence of potential neuron and network compensatory responses to KCNT1 GOF

In a healthy brain, there are multiple homeostatic compensation mechanisms to maintain neuronal and network activity at their proper physiological levels (*Davis, 2013*; *Debanne et al., 2019*; *Debanne and Russier, 2019*; *Marder and Goaillard, 2006*; *Turrigiano, 2011*; *Turrigiano, 2012*; *Yang and Prescott, 2023*), whereas in a brain with epilepsy-causing genetic mutations or injury, it is thought that failure of homeostatic compensation can result in hypersynchronous, hyperexcitable networks and, ultimately, seizures (*Issa et al., 2023*; *Lignani et al., 2020*; *Staley, 2015*). As YH-HET mice have only been observed to have infrequent seizures and YH-HET networks appear largely intact, it is plausible that there are adaptive or homeostatic mechanisms downstream of the YH variant that limit KCNT1 GOF effects largely to those observed at the current and neuron level. These regulatory mechanisms may include alterations in ion currents and/or synaptic input to restore proper AP shape or neuronal excitability, or alterations in chemical and/or electrical connectivity to stabilize neural networks.

Neuronal excitability is regulated by the inward and outward flow of opposing currents through sodium and potassium channels; thus, it is not surprising that mutations affecting these channels can lead to compensatory up- or downregulation of other ion channels or currents to maintain excitability and proper neuron function. In this study, our in silico electrophysiology predictions of YH-HET PV neuron excitability were opposite to those of our in vitro electrophysiology data; thus, we hypothesized that there may be a compensatory increase in an ion current that counteracts the increased $K_{Na}$ current selectively in YH-HET PV neurons. Accordingly, we identified a subthreshold increase in the persistent $Na^+$ current, across a similar voltage range to the increased $K_{Na}$ current, in YH-HET PV, but not SST, neurons. Furthermore, in silico approaches indicated that this increased $I_{NaP}$ was sufficient to overcome the effects of KCNT1 GOF and cause an overall increase in AP generation in YH-HET PV neurons, which aligned with our experimental data. Similar compensatory mechanisms have been identified in other models such as the downregulation of multiple potassium channels in a sodium channel (*Scn2a*) LOF mouse model of epilepsy, resulting in neuronal hyperexcitability (*Zhang et al., 2021*), or the upregulation of $Na^+$ currents in response to increased potassium channel activity in a developing *Xenopus* neural circuit, resulting in restored intrinsic excitability and network stability (*Pratt and Aizenman, 2007*). Notably, another KCNT1 GOF model (*Kcnt1*-R455H) showed upregulation of a sodium channel ($Na_v1.6$), accompanied by an increase in transient and persistent sodium currents, in *Kcnt1*-R455H cortical neurons, but the effects of Nav1.6 upregulation were not investigated (*Wu et al., 2024*).

As the persistent sodium current has been shown to act as a source of cytoplasmic sodium ions for KCNT1 channel activation in some neuron types (*Hage and Salkoff, 2012*), one might expect that the compensatory increase in $I_{NaP}$ in YH-HET PV neurons would further increase, rather than counteract, $K_{Na}$ currents. Unfortunately, there is insufficient information on the relative locations of the $I_{NaP}$ and KCNT1 channels, as well as the kinetics of sodium transfer to KCNT1 channels, among cortical neuron subtypes, and even less is known in the context of KCNT1 GOF neurons; thus, it is difficult to predict how alterations in one of these currents may affect the other. One plausible reason that increased $I_{NaP}$ would not alter $K_{Na}$ currents in YH-HET PV neurons is that the particular sodium channels that are responsible for the increased $I_{NaP}$ are not located within close proximity to the KCNT1 channels. Moreover, homeostatic mechanisms that modify the length and/or location of the sodium channel-enriched axon initial segment (AIS) in neurons in response to altered excitability are well described (*Grubb and Burrone, 2010*; *Kuba et al., 2010*); thus, it is possible that in YH-HET PV neurons, the length or location of the AIS is altered, leading to uncoupling of the sodium channels that are responsible for the increased $I_{NaP}$ to the KCNT1 channels. Future studies will aim to further investigate potential mechanisms of neuron-type-specific alterations in $Na_P$ and $K_{Na}$ currents downstream of KCNT1 GOF.

In addition to the KCNT1 GOF current-mediated impairments in SST neuron excitability, we identified several potential compensatory YH-HET SST synaptic and network alterations. For instance, there was an increased frequency of sEPSCs onto YH-HET SST neurons, suggestive of a compensatory increase in excitatory input to counteract their reduced intrinsic excitability and increase their AP firing, as has been observed previously in hypoexcitable SST neurons in another epilepsy model (*Halabisky et al., 2010*). There was also an increase in chemical and electrical coupling between YH-HET SST neurons. Although SST neurons normally show a relatively low rate of homotypic chemical coupling, particularly in the adult cortex, more than half of nearby SST neuron pairs are coupled electrically

(*Amitai et al., 2002*; *Gibson et al., 1999*; *Gibson et al., 2005*; *Hu and Agmon, 2015*; *Urban-Ciecko and Barth, 2016*). This coupling is thought to be important for fine tuning neural circuits by regulating such processes as synchronous AP firing, synaptic integration, and network rhythmicity (*Alcamí and Pereda, 2019*; *Connors, 2017*). The observed increase in electrical coupling should not only contribute to the reduced intrinsic excitability properties of individual YH-HET SST neurons, increasing their capacitance and decreasing their input resistance, but may also act as a homeostatic mechanism to restore synchronicity and stability to YH-HET SST networks (*Lane et al., 2016*; *Parker et al., 2009*).

Taken together, how might the YH-HET variant-induced alterations to SST and PV neurons, both direct and adaptive, ultimately contribute to neural network behavior and seizure propensity in YH-HET mice? The paucity of effects on synaptic activity and connectivity in YH-HET networks, relative to those of YH-HOM, suggests that the current- and neuron-level effects of the YH-HET variant are below some threshold required to induce broader network hyperexcitability and seizures. Moreover, as described above, several of the proposed homeostatic mechanisms, such as increased excitatory input onto, and connectivity between, SST neurons and increased $I_{NaP}$ in PV neurons, should theoretically act to suppress pathological network activity in YH HET mice. However, predicting the downstream effects of altered SST and PV intrinsic excitability is not straightforward, particularly with the emerging appreciation of both anti- and pro-epileptic roles for these neuron types in seizure initiation and maintenance (*Dudok et al., 2022*; *Magloire et al., 2019*). Furthermore, studies using genetic mouse models of DEEs have identified seemingly paradoxical, and sometimes ephemeral, roles of interneurons in disease progression. For instance, in an *SCN8A* GOF DEE mouse model, increased $I_{NaP}$ causes initial SST hyperexcitability followed by depolarization block, and in parallel, chemogenetic activation of WT SST neurons mimics these effects, leading to seizure generation (*Wengert et al., 2021*). Similarly, in a *KCNQ2/3* LOF DEE model, PV-specific *Kcnq2/3* deletion causes early PV hyperexcitability that is thought to increase seizure susceptibility via homeostatic potentiation of excitatory transmission (*Soh et al., 2018*). Lastly, in an *Scn1a*$^{+/-}$ mouse model of Dravet syndrome, PV neurons are transiently hypoexcitable during development, whereas later they show an increased firing rate, both interictally and at seizure onset, and a decrease in synchrony in the transition to seizure (*Favero et al., 2018*; *Tran et al., 2020*). These studies illustrate the challenges in connecting altered interneuron intrinsic excitability with network behavioral outcomes not only because of the contradictory nature of some of these connections but also because of the often transient nature of interneuron phenotypes during disease progression. Thus, future studies should aim to assess neuron-type-specific effects of the YH variant at different developmental time points in both YH-HET mice, with rare seizures, and YH-HOM mice, with frequent seizures, to better understand the contributions of interneuron dysfunction to network pathology and seizure generation in KCNT1 GOF-related epilepsy.

## Limitations and conclusions

The in vitro and in silico nature of this study are limitations. Although the cultured VIP, SST, and PV neurons showed characteristic electrophysiological properties, it is unknown how much overlap there is between the in vitro and in vivo populations. For instance, it is possible that our culture conditions select for more mature neurons of each type or preferentially support the survival of subpopulations; thus, we may be assessing KCNT1 GOF effects on a small portion of a given subtype. However, an advantage of the in vitro prep is that the compensatory alterations we observe are not likely consequences of seizure activity, which has been shown to alter interneuron properties in epilepsy models. Furthermore, the in silico neurons we used to model each neuron type were designed to accurately reflect the morphological and electrophysiological properties of each subtype, but they may lack more detailed features of ion channels, such as post-translational modifications and subcellular localizations, that can have important functional effects. Our KCNT1 model conductance is also hampered by an incomplete understanding of the relationship between Na$^+$ influx, membrane voltage, and channel gating in neurons.

Moreover, our K$^+$ current measures using *Kcnt1*$^{-/-}$; *Kcnt2*$^{-/-}$ double knockout neurons clearly showed the presence of an outward VU170-sensitive current across positive potentials, indicating non-specific activity of this small molecule, potentially to the ERG K$^+$ channel (*Spitznagel et al., 2020*). This off-target effect of VU170 may contribute to the overall increase in outward K$^+$ currents seen in our VU170- vs. TTX-subtracted currents in all neuron types across positive potentials, and it limits the interpretation of the VU170-sensitive K$_{Na}$ current data within this range. Finally, although it is well

known that temperature changes can alter neuron physiology, the electrophysiological recordings in this study were performed at room temperature rather than physiological temperature. In this study, we further assessed the effect of temperature (22 vs. 35°C) on rheobase, AP amplitude, and maximal AP firing frequency among WT and YH-HET PV neurons and showed that both groups responded similarly. Previously, we also showed that the effects of the homozygous YH variant on glutamatergic and NFS GABAergic neuron excitability were highly similar between cultured recordings at room temperature (~22°C) and slice recordings at 32°C. Taken together, these data suggest that the reported neurophysiological phenotypes downstream of the YH variant are likely not temperature dependent.

Despite the potential limitations, this study provides three major findings that advance our understanding of the relationship between ion channel function and disease. First, heterozygous *KCNT1* expression causes changes in current flow and neuron excitability that are qualitatively similar to homozygous expression, but of a lower magnitude; however, the synaptic activity and connectivity changes are different, which likely leads to the discrepancy in seizure incidence. This finding suggests a non-linear relationship between GOF, at the level of the ion channel current, and disease severity. Second, the same *KCNT1* variant can produce opposite effects on neuron excitability in closely related GABAergic neuron subtypes, and these opposite effects are likely due to compensation in one neuron type (PV) that is absent in the other (SST). Previous studies have observed opposite effects on neuronal excitability due to $Na^+$ channel variants, but the underlying mechanism, when explored, was proposed to be intrinsic differences in ion channel expression, not compensation. Third, we observed increased synaptic and gap junction connectivity among SST neurons, demonstrating that the effects of KCNT1 GOF extend to structural alterations as well. The effects of this increased connectivity, especially by gap junctions, on neuronal excitability and network behavior offers an exciting avenue for future research. Finally, it will be critical to determine whether these alterations downstream of KCNT1 GOF potentiate or attenuate network pathology and seizure activity, thus providing a better understanding of disease mechanisms and prompting novel therapeutic design.

# Materials and methods

**Key resources table**

| Reagent type (species) or resource | Designation | Source or reference | Identifiers | Additional information |
|---|---|---|---|---|
| Genetic reagent (*Mus musculus*) | *Kcnt1*Y777H knockin mice | The Jackson Laboratory | Jackson Labs stock: 028830 | |
| Genetic reagent (*Mus musculus*) | *Sst*-IRES-Cre mice | The Jackson Laboratory | Jackson Labs stock: 013044 | |
| Genetic reagent (*Mus musculus*) | *Vip*-IRES-Cre mice | The Jackson Laboratory | Jackson Labs stock: 010908 | |
| Genetic reagent (*Mus musculus*) | *Pvalb*-IRES-Cre mice | The Jackson Laboratory | Jackson Labs stock: 017320 | |
| Genetic reagent (*Mus musculus*) | GIN mice | The Jackson Laboratory | Jackson Labs stock: 003718 | |
| Strain, strain background (*Mus musculus*) | C57BL/6J | The Jackson Laboratory | Jackson Labs stock: 000644 | |
| Sequence-based reagent | Kcnt1YH_F | This paper | IDT DNA | CTAGGGCTG CAAACACAACA |
| Sequence-based reagent | Kcnt1YH_R | This paper | IDT DNA | TCAAGCAGCA ACACGATAGG |
| Sequence-based reagent | Cre_F | This paper | IDT DNA | TCGCGATTATCT TCTATATCTTCAG |
| Sequence-based reagent | Cre_R | This paper | IDT DNA | GCTCGACCAG TTTAGTTACCC |
| Recombinant DNA reagent | AAV8-*CaMKIIa*-GFP | UNC Vector Core | | |
| Recombinant DNA reagent | AAV9-*hSyn*-DIO-mCherry | Addgene | 50459-AAV9 | |

*Continued on next page*

Continued

| Reagent type (species) or resource | Designation | Source or reference | Identifiers | Additional information |
|---|---|---|---|---|
| Chemical compound, drug | NBQX disodium salt | Tocris | 1044 | |
| Chemical compound, drug | Bicuculline methiodide | hello bio | HB0893 | |
| Chemical compound, drug | Tetrodotoxin citrate | abcam | ab120055 | |
| Software, algorithm | pClamp, Clampex 10.3 or 10.5, Clampfit 11.2 | Molecular Devices | RRID:SCR_011323 | |
| Software, algorithm | Axograph X | Axograph Scientific | RRID:SCR_014284 | |
| Software, algorithm | Blue Brain Project | EPFL | RRID:SCR_002994 | |
| Software, algorithm | NEURON 8.0 | The NEURON Simulator | RRID:SCR_005393 | |
| Software, algorithm | μManager 2.0-β | μManager | RRID:SCR_000415 | |
| Software, algorithm | Fiji | ImageJ, NIH | RRID:SCR_002285 | |
| Software, algorithm | Prism 10 | GraphPad Prism | RRID: SCR_002798 | |
| Software, algorithm | SPSS 28.0 | IBM | RRID:SCR_002865 | |
| Antibody | Neu-N | Synaptic Systems | 266-004 | 1:500 |
| Antibody | Alexa Fluor 647 | Invitrogen | A21450 | 1:1000 |
| Other | Fluoromount-G Mounting Medium, with DAPI | Thermo Fisher Scientific | 00-4959-52 | See *Immunostaining, Imaging, and Quantification* section for more information |

## Mice

Mice were bred, and mouse procedures were conducted, in compliance with the National Institutes of Health (NIH) Guidelines for the Care and Use of Laboratory Animals and were approved by the Institutional Animal Care and Use Committee at the University of Vermont (animal protocol numbers: 16-001, 19-034, and X9-022). Mice were maintained in ventilated cages at controlled temperature (22–23°C), humidity ~60%, and 12 hr light: 12 hr dark cycles (lights on at 7:00 AM, off 7:00 PM). Mice had access to regular chow and water, ad libitum. For all experiments, male and female littermates were used for each genotype. The ages of the mice for each experiment are indicated in the following relevant sections.

Mouse lines and strains used for these studies include: $Kcnt1^{Y777H}$ knockin in the C57BL/6NJ (B6NJ) strain ($Kcnt1^{em1(Y777H)Frk}$; Jackson Labs stock: 028830), Sst-IRES-Cre ($Sst^{tm2.1(cre)Zjh}$/J; Jackson Labs stock: 013044), Vip-IRES-Cre ($Vip^{tm1(cre)Zjh}$/J; Jackson Labs stock: 010908), Pvalb-IRES-Cre (B6.129P2-$Pvalb^{tm1(cre)Arbr}$/J; Jackson Lab stock: 017320), $Kcnt1^{-/-}$; $Kcnt2^{-/-}$ double knockout (a generous gift from Dr. Christopher Lingle at WashU), GIN (FVB-Tg(GadGFP)45704Swn/J; Jackson Lab stock: 003718), and C57BL/6J (Jackson Labs stock: 000664) mice.

$Kcnt1^{Y777H}$ mice were genotyped using PCR amplification primers ($Kcnt1$ forward primer: 5'-CTAG GGCTGCAAACACAACA-3'; $Kcnt1$ reverse primer: 5'-TCAAGCAGCAACACGATAGG-3') with standard thermocycler amplification conditions, and the annealing temperature set at 58°C. Following amplification, a restriction cut was performed with the enzyme NlaIII to distinguish mutant (127 and 44 bp products after cut) from WT alleles (171 bp product). Progeny of the $Kcnt1^{Y777H}$ mice crossed to each Cre mouse line were genotyped for the presence of a Cre transgene using a forward primer (5'-TCGCGATTATCTTCTATATCTTCAG-3') and reverse primer (5'-GCTCGACCAGTTTAGTTACCC-3'), resulting in a 455-bp product.

## Primary astrocyte feeder layer culture

Astrocyte feeder layers were generated to support the growth and maintenance of primary neurons. Briefly, cortices were dissected from P0 to P1 WT C57BL/6J mice (Jackson Labs stock: 000664) of either sex. The cortices were incubated in 0.05% trypsin–EDTA (Gibco) for 15 min at 37°C in a Thermomixer (Eppendorf) with gentle agitation (800 rpm). Then, the cortices were mechanically dissociated

with a 1-ml pipette tip, and the cells were plated into T-75 flasks containing filter-sterilized astrocyte media [Dulbecco's Modified Eagle Medium (DMEM) supplemented with glutamine (Gibco), 10% fetal bovine serum (GE Healthcare), 1× MITO +Serum Extender (Corning), and 0.2× penicillin/streptomycin (Gibco)]. After the astrocytes reached confluency, they were washed with phosphate-buffered saline (PBS; Gibco) and incubated for 5 min in 0.05% trypsin–EDTA at 37°C, washed, and then resuspended in astrocyte media. Astrocytes were added to 6-well plates containing 25 mm coverslips precoated with coating mixture [0.7 mg/ml collagen I (Corning) and 0.1 mg/ml poly-d-lysine (Sigma) in 10 mM acetic acid].

## Primary cortical neuron culture

For the primary neuron culture, the dorsomedial cortices from P0 to P1 WT and $Kcnt1^{m/+}$ mice of both sexes were dissected in cold Hank's Balanced Salt Solution (HBSS; Gibco). The tissue was then digested with papain (Worthington) for 60–75 min and treated with inactivating solution (Worthington) for 10 min, both while shaking at 800 rpm at 37°C in a Thermomixer. The cells were then mechanically dissociated and counted. The dissociated cells were added at 200,000 cells/well to 6-well plates containing astrocyte-coated coverslips in filter-sterilized NBA plus [Neurobasal-A medium (Gibco) supplemented with 1× Glutamax (Gibco), 1× B27 (Invitrogen), and 0.2× penicillin/streptomycin (Gibco)]. After plating (12–24 hr), approximately $4 \times 10^{10}$ genome copies (GC) of AAV8-$CaMKIIa$-GFP (UNC Vector Core) were added to each well. For experiments to assess KCNT1 effects on GABAergic subtypes, approximately $4 \times 10^{10}$ GC of AAV9-$hSyn$-DIO-mCherry (Addgene) were also added to each well to mark Cre-expressing neurons. Every 3–4 days, 20–40% of the media was replaced with fresh NBA plus.

## Electrophysiology

Whole-cell recordings were performed with patch-clamp amplifiers (MultiClamp 700B; Molecular Devices) under the control of Clampex 10.3 or 10.5 (Molecular Devices, pClamp, RRID:SCR_011323). Data were acquired at 20 kHz and low-pass filtered at 6 kHz. The series resistance was compensated at 70%, and only cells with series resistances maintained at less than 15 MΩ were analyzed. Patch electrodes were pulled from 1.5 mm o.d. thin-walled glass capillaries (Sutter Instruments) in five stages on a micropipette puller (model P-97; Sutter Instruments). Internal solution contained the following: 136 mM K-gluconate, 17.8 mM HEPES, 1 mM EGTA, 0.6 mM MgCl₂, 4 mM ATP, 0.3 mM GTP, 12 mM creatine phosphate, and 50 U/ml phosphocreatine kinase. Alternatively, internal solution contained: 136 mm KCl, 17.8 mm HEPES, 1 mm EGTA, 0.6 mm MgCl₂, 4 mm ATP, 0.3 mm GTP, 12 mm creatine phosphate, and 50 U/ml phosphocreatine kinase. The pipette resistance was between 2 and 4 MΩ. Standard extracellular solution contained the following (in mM): 140 NaCl, 2.4 KCl, 10 HEPES, 10 glucose, 4 MgCl₂, and 2 CaCl₂ (pH 7.3, 305 mOsm). All experiments were performed at room temperature (22–23°C), with the exception of the current-clamp experiments to compare the effects of temperature on neuronal physiology (***Figure 4—figure supplement 3***). For these recordings, all current-clamp parameters were measured first at room temperature, then the temperature of the external solution was heated to 35°C using a ThermoClamp Inline Heater (ThermoClamp-3; AutoMate Scientific), and the measurements were repeated. Whole-cell recordings were performed on cortical neurons from control and mutant groups in parallel on the same day (day 13–17 in vitro). All experiments were performed by two independent investigators blinded to the genotypes.

For current-clamp experiments, the intrinsic electrophysiological properties of neurons were tested by injecting 500 ms square current pulses incrementing in 20 pA steps, starting at −100 pA. Resting membrane potential ($V_m$) was calculated from a 50-ms average before current injection. The membrane time constant ($\tau$) was calculated from an exponential fit of current stimulus offset. Input resistance ($R_{In}$) was calculated from the steady state of the voltage responses to the hyperpolarizing current steps. Membrane capacitance was calculated by dividing the time constant by the input resistance. APs were evoked with 0.5 s, 20 pA depolarizing current steps. Rheobase was defined as the minimum current required to evoke an AP during the 500 ms of sustained somatic current injections. AP threshold was defined as the membrane potential at the inflection point of the rising phase of the AP. AP amplitude was defined as the difference in membrane potential between the AP peak and threshold, and the AHP was the difference between the AP threshold and the lowest $V_m$ value within 50ms. The AP half-width was defined as the width of the AP at half-maximal amplitude. To obtain the neuron's maximum

firing frequency, depolarizing currents in 20 pA steps were injected until the number of APs per stimulus reached a plateau phase. The membrane potential values were not corrected for the liquid junction potential. GABAergic neurons were classified as FS if their maximum mean firing rate reached above 60 Hz and their AP half-widths increased by less than 25% during 1 s of sustained firing (*Avermann et al., 2012*; *Casale et al., 2015*). All others were considered NFS.

For voltage-clamp experiments to measure synaptic currents, neurons were held at −70 mV, except for evoked IPSC measurements, for which neurons were held at 0 mV. AP-evoked EPSCs were triggered by a 2-ms somatic depolarization to 0 mV. The shape of the evoked response, the reversal potential and the effect of receptor antagonists [10 µM NBQX (Tocris Bioscience) or 20 µM bicuculline (BIC, hello bio)] were analyzed to verify the glutamatergic or GABAergic identities of the currents. Neurons were stimulated at 0.1 Hz in standard external solution to measure basal-evoked synaptic responses. Electrophysiology data were analyzed offline with AxoGraph X software (AxoGraph Scientific, RRID:SCR_014284). Spontaneous synaptic potentials were recorded in control solution with either NBQX or BIC to isolate EPSCs or IPSCs, respectively. Data were filtered at 1 kHz and analyzed using template-based miniature event detection algorithms implemented in the AxoGraph X. The threshold for detection was set at three times the baseline SD from a template of 0.5 ms rise time and 3 ms decay. The E/I ratio was calculated as the product of the sEPSC frequency and charge over the sum of the sEPSC frequency and charge and the product of the sIPSC frequency and charge.

## $K_{Na}$ and KCNT1 current measurements

For voltage-clamp experiments to measure the sodium-activated $K^+$ current or KCNT1-mediated current, neurons were held at −70 mV and given 1 s voltage pulses in 10 mV steps over a range of −80 to +50 mV. Recordings were obtained for each cell in standard extracellular solution or extracellular solution containing 0.5 µM TTX or 10 µM VU170. TTX or VU170 was applied directly on the recorded neuron with a custom-built fast flow perfusion system capable of complete solution exchange in less than 1 s to minimize the time interval between control and drug recordings and current rundown. Current traces from the TTX or VU170 solution were subtracted from the current traces obtained from the standard solution. The difference current over the 100ms at the end of the voltage pulse was considered the steady-state $K_{Na}$ current. The same protocol was used for outside-out patch experiments.

## $I_{NaP}$ current measurements

For voltage-clamp experiments to measure the persistent $Na^+$ current, neurons were held at −80 mV and given a 5-s voltage ramp at 20 mV/s. The ramp was repeated every 10 s. Recordings were obtained for each cell in standard extracellular solution or extracellular solution containing 0.5 µM TTX. TTX was applied directly on the recorded neuron with a custom-built fast flow perfusion system capable of complete solution exchange in less than 1 s to minimize the time interval between control and TTX recordings. The first current trace after TTX application was subtracted from the averaged current traces obtained from the standard solution. At this time point, the effect of TTX on the $Na^+$-activated $K^+$ current is minimal, while the persistent $Na^+$ current is maximally inhibited. This allowed us to isolate the persistent $Na^+$ current from $K_{Na}$ and obtain a more accurate estimate of changes in $I_{NaP}$.

## Neuron modeling

Model neurons (~10 per class) representing excitatory (Pyramidal Cell, PC), SST (Martinotti Cell, MC), PV (Large Basket Cell, LBC), and VIP neurons (Bipolar Cell, BP) were downloaded from the EPFL/Blue Brain project (https://bbp.epfl.ch/nmc-portal/downloads.html), and implemented in the NEURON environment. Because no KCNT1-like conductance was present in the original models, a Hodgkin-Huxley KCNT1 model conductance was created with both $Na^+$- and voltage-dependent gates and inserted into the dendritic, somatic, and axonal compartments of model neurons (0.03 S/cm$^2$). The $Na^+$ dependence was modeled after Bischoff et al. ($EC_{50}$ = 40 mM, slope 3.5), and the voltage dependence was modeled after our own experimental data (*Figure 5*), corrected to account for the calculated liquid junction potential in the experimental data. To verify that this approach reproduced experimental data, voltage step protocols were run in NEURON with 10 and 0 mM internal $Na^+$. '$K_{Na}$' currents were obtained by subtraction, and the amplitude and kinetics of the resulting currents were verified to be similar to experimental values. To model the effects of KCNT1 GOF,

the EC$_{50}$ for Na$^+$ was decreased to either 35 or 30 mM, because the Y796H variant was previously shown to increase the Na$^+$ sensitivity of the channel. These changes resulted in moderate increases in the current above WT levels (*Figure 6B*), consistent with the experimental data (*Figures 4 and 5*). Modeling data were generated by running current clamp protocols in NEURON for each of the model neurons and analyzed in the same way as experimental data. For modeling the increase in $I_{NaP}$, the control $I_{NaP}$ conductance was set to 0.001 times the conductance of the transient Na$^+$ current in each compartment. This produced $I_{NaP}$ currents approximately 1% of the peak of the transient Na$^+$ current. KCNT1 GOF was modeled by decreasing the Na$^+$ EC$_{50}$ to 30 mM and the $I_{NaP}$ increase was modeled by increasing its conductance twofold, in accordance with experimental data.

## Immunostaining, imaging, and quantification

For live imaging of GABAergic subtypes in vitro, cortical neurons isolated and cultured from VIP-Cre, SST-Cre, and PV-Cre pups, were infected with AAV-*CamKIIa*-GFP and AAV-*hSyn*-DIO-mCherry viruses at DIV 1 to mark glutamatergic and Cre-expressing, GABAergic neurons, respectively. Images were captured from live cultures at DIV 14 (VIP-Cre and SST-Cre) or DIV 16 (PV-Cre) using an inverted Olympus IX73 epifluorescent microscope with an Olympus PlanFluor ×10 objective lens and an Andor Zyla sCMOS 4.2 camera controlled by μManager 2.0-β software (*Edelstein et al., 2014*). The same acquisition parameters were used for each image and each group.

For soma size measurements, DIV 16 cultured SST-Cre neurons on coverslips were washed three times with prewarmed PBS, fixed with 4% paraformaldehyde in PBS for 30 min at room temperature, and then washed again three times with PBS. The fixed neurons were then incubated in blocking buffer (10% normal goat serum and 0.1% Triton-100 in 1× PBS) for 1 hr at room temperature, followed by incubation with a Neu-N primary antibody (Synaptic Systems 266-004) diluted in blocking buffer (1:500) overnight at 4°C. The next day, the neurons were washed three times with PBS for 10–15 min, incubated with goat anti-guinea pig, Alexa Fluor 647 secondary antibody (Invitrogen A21450) in blocking buffer (1:1000) for 1 hr at room temperature, and washed again three times with PBS for 10–15 min. The coverslips were then flipped and mounted on glass slides using Fluoromount G with DAPI (Thermo Fisher Scientific). Images (1024 × 1024 pixels) of Neu-N to mark all neurons, *CamKI-Ia*-GFP to mark glutamatergic neurons, and *hSyn*-DIO-mCherry to mark Cre-expressing neurons, were obtained using a DeltaVision Restoration Microscopy System (Applied Precision/GE Life Sciences) with an inverted Olympus IX70 microscope with a ×20 oil objective, SoftWoRx software, and a Cool-SNAP-HQ charge-coupled device digital camera (Photometrics). Image exposure times and settings were kept the same between groups in a culture and were optimized to ensure that there were no saturated pixels. To analyze soma size, regions of interest were drawn around cell bodies using the mCherry channel with Fiji software (*Schindelin et al., 2012*), and then the cell body areas were measured for each mCherry$^+$ neuron imaged.

## Mouse cortex slice preparation

Cortical slices were prepared from P21 to P30 WT and *Kcnt1*$^{Y777H}$ heterozygous pups of both sexes that were the progeny of *Kcnt1*$^{Y777H}$ heterozygous mice crossed to homozygous GIN mice (Jackson Labs stock: 003718), which express GFP in a subset of SST-positive neurons (*Ma et al., 2006*; *Oliva et al., 2000*), following a modified protocol (*Ting et al., 2014*). Briefly, mice were deeply anesthetized with isoflurane and transcardially perfused with 25–30 ml of ice-cold N-methyl-D-glucamine (NMDG)-based artificial cerbrospinal fluid (aCSF) (in mM: 93 NMDG, 2.5 KCl, 1.2 NaH$_2$PO$_4$, 30 NaHCO$_3$, 20 HEPES, 25 D-glucose, 5 Na-ascorbate, 2 thiourea, 3 Na-pyruvate, 12 *N*-acetyl-L-cysteine, 10 MgSO$_4$, 0.5 CaCl$_2$, with pH adjusted to 7.3–7.4 with 10 N HCl) that was saturated with carbogen 95% O$_2$, 5% CO$_2$ (pH 7.3–7.4). The mice were rapidly decapitated, and the heads were submerged in ice-cold NMDG aCSF bubbled with carbogen. After extraction from the skull, the cortex was blocked and 250 μm coronal sections were cut with a vibrating microslicer (Leica VT100S, Leica Biosystems) while submerged in the same solution. Slices were first moved to NMDG aCSF at 32–34°C for a short-term recovery (≤15 min) before being transferred into HEPES-based holding aCSF (in mM: 92 NaCl, 2.5 KCl, 1.2 NaH$_2$PO$_4$, 30 NaHCO$_3$, 20 HEPES, 25 D-glucose, 5 Na-ascorbate, 2 thiourea, 3 Na-pyruvate, 12 *N*-acetyl-L-cysteine, 2 MgSO$_4$, 2 CaCl$_2$, bubbled with carbogen, with pH adjusted to 7.3–7.4 with KOH or HCl as necessary). The slices were held in this solution until use.

## Ex vivo slice electrophysiology

For recording, the cortical slices were placed in a chamber mounted on a fluorescence microscope (Zeiss Axio Examiner.D1) and perfused with normal Ringer's solution containing (in mM): 130 NaCl, 3 KCl, 2 $MgCl_2$, 2 $CaCl_2$, 1.25 $NaH_2PO_4$, 26 $NaHCO_3$, and 10 D-glucose, bubbled with carbogen. Patch electrodes were pulled from 1.5 mm o.d. thin-walled glass capillaries (Garner Glass Company) in two stages on a micropipette puller (model P-1000; Sutter Instruments). Electrophysiological recordings were obtained from GFP-labeled neurons in layer 2/3 of either motor or somatosensory cortex using pClamp, an MultiClamp 700B amplifier, and a Digidata 1440A analog-to-digital converter (Molecular Devices). Data were acquired at 20 kHz and low-pass filtered at 10 kHz. Internal solution contained the following (in mM): 17.5 KCl, 112.5 K-gluconate, 1.5 NaCl, 5 $Na_2$-phosphocreatine, 1 $MgCl_2$, 10 HEPES, 0.2 EGTA, 3 Mg-ATP, 0.3 GTP-Tris (pH 7.2, with KOH). The pipette resistance was between 3 and 5 MΩ. All experiments were performed at room temperature (22–23°C).

The intrinsic electrophysiological properties of neurons were tested using current-clamp recording under whole-cell configuration by injecting 1 s square current pulses in 5 pA steps, starting at −30 pA. All recordings were saved in the computer for off-line analysis using Clampfit 11.2 or Axograph X software. The membrane time constant ($\tau$) was calculated from an exponential fit of current stimulus offset. Input resistance ($R_{in}$) was calculated from the steady state of the voltage responses to the hyperpolarizing current steps. Membrane capacitance was calculated by dividing the time constant by the input resistance. AP frequency was counted on each depolarizing current step and plotted as a function of injection current. Rheobase was defined as the minimum current required to evoke an AP during the 1 s of sustained somatic current injections. AP threshold was defined as the membrane potential at the inflection point of the rising phase of the AP, and the AHP was the difference between the AP threshold and the lowest $V_m$ value within 50 ms. The membrane potential values were not corrected for the liquid junction potential.

## Data presentation and statistical analysis

Prism 10 (GraphPad Prism, RRID:SCR_002798) was used to create all graphs. To test for statistical significance for all whole-cell electrophysiology experiments, we used generalized linear mixed models in SPSS [28.0 Chicago, III (IBM, RRID:SCR_002865)], which allows for within-subject correlations and the specification of the most appropriate distribution for the data. Because neurons and animals from the same culture or animal are not independent measurements, culture or litter was used as the subject variable, and animals, neurons, and voltage steps were considered within-subject measurements. All data distributions were assessed with the Shapiro–Wilk test. Datasets that were significantly different from the normal distribution ($p < 0.05$) were fit with models using the gamma distribution and a log link function, except for synaptic connection probability, which was fit with the binomial distribution and probit link. Normal datasets were fit with models using a linear distribution and identity link. We used the model-based estimator for the covariance matrix and goodness of fit was determined using the corrected quasi likelihood under independence model criterion and by the visual assessment of residuals. All values reported in the text, figures, and tables are estimated marginal means ± standard error, except where noted.

## Acknowledgements

This work was supported by NIH/NINDS grants R01NS110945 (MCW), R01NS130042 (MCW), R01NS031348 (WNF), R33NS109521 (CDW and KAE), and U54OD020351 (WNF, The Jackson Laboratory Center for Precision Genetics). The authors would like to acknowledge the support of the Microscopy Imaging Center (RRID# SCR_018821) and the Cellular & Molecular Core at the University of Vermont. The authors would like to thank Dr. Christopher Lingle of Washington University of St. Louis for providing the *Kcnt1*$^{-/-}$; *Kcnt2*$^{-/-}$ double knockout mice to test the specificity of the VU170 KCNT1 inhibitor.

## Additional information

### Funding

| Funder | Grant reference number | Author |
|---|---|---|
| NIH/NINDS | R01NS130042 | Matthew C Weston |
| NIH/NINDS | R01NS031348 | Wayne Frankel |
| NIH/NINDS | R33NS109521 | C David Weaver<br>Kyle Emmitte |
| NIH/NINDS | U54OD020351 | Wayne Frankel |
| NIH/NINDS | R01 NS110945 | Matthew C Weston |

The funders had no role in study design, data collection and interpretation, or the decision to submit the work for publication.

### Author contributions

Amy N Shore, Conceptualization, Data curation, Formal analysis, Supervision, Investigation, Visualization, Methodology, Writing – original draft, Project administration, Writing – review and editing; Keyong Li, Mona Safari, Formal analysis, Investigation; Alshaima'a M Qunies, Brittany D Spitznagel, C David Weaver, Kyle Emmitte, Wayne Frankel, Resources; Matthew C Weston, Conceptualization, Data curation, Formal analysis, Supervision, Funding acquisition, Investigation, Methodology, Writing – original draft, Project administration, Writing – review and editing

### Author ORCIDs

Wayne Frankel ⓘ https://orcid.org/0000-0003-2241-5314
Matthew C Weston ⓘ https://orcid.org/0000-0001-5558-7070

### Ethics

This study was performed in strict accordance with the recommendations in the Guide for the Care and Use of Laboratory Animals of the National Institutes of Health. All surgery was performed under isofluorane anesthesia, and every effort was made to minimize suffering. All mice were bred, and procedures were conducted at the Fralin Biomedical Research Institute at VTC, or at the University of Vermont. Each institution is fully accredited by the Association for Assessment and Accreditation of Laboratory Animal Care, and all protocols were approved by their respective Institutional Animal Care and Use Committees. All experiments were performed in accordance with respective state and federal Animal Welfare Acts and the policies of the Public Health Service. The animal protocol numbers for Virginia Tech were 22-198 and 22-130, and the University of Vermont were 16-001 and 19-034.

Reviewer #1 (Public Review): https://doi.org/10.7554/eLife.92915.4.sa1
Reviewer #2 (Public Review): https://doi.org/10.7554/eLife.92915.4.sa2
Reviewer #3 (Public Review): https://doi.org/10.7554/eLife.92915.4.sa3
Author response https://doi.org/10.7554/eLife.92915.4.sa4

## Additional files

### Supplementary files

• Supplementary file 1. Table shows values (mean ± SEM) of all electrophysiological parameters measured from wildtype (WT) and YH-HET glutamatergic, and fast spiking (FS) and non-fast spiking (NFS) GABAergic, neurons. For an explanation of the parameters, see Methods. $R_{in}$ = input resistance, Tau = membrane time constant, Rheo = rheobase current, $C_m$ = membrane capacitance, AP = action potential, thresh = threshold, amp = amplitude, hw = half width, AHP = afterhyperpolarization, and mfr = maximum firing rate. Values shown are estimated marginal means ±the standard error as determined by implementing a generalized linear mixed model. p-values less than 0.05 are in red, bold type. For each subgroup, $N$ values are the number of mouse pups, and $n$ values are the number of neurons.

• Supplementary file 2. Electrophysiological parameters of current clamp recordings from

neuronal cultures of three major GABAergic subtypes. Table shows values (mean ± SEM) of all electrophysiological parameters measured from WT and YH-HET, SST, VIP, and PV GABAergic neurons. For an explanation of the parameters, see Methods. $R_{in}$ = input resistance, Tau = membrane time constant, Rheo = rheobase current, $C_m$ = membrane capacitance, AP = action potential, thresh = threshold, amp = amplitude, hw = half width, AHP = afterhyperpolarization, and mfr = maximum firing rate. Values shown are estimated marginal means ± the standard error as determined by implementing a generalized linear mixed model. p values less than 0.05 are in red, bold type. For each subgroup, N values are the number of mouse pups, and n values are the number of neurons.

• Supplementary file 3. Electrophysiological parameters using compartmental model neurons. Table shows values (mean ± SEM) of all electrophysiological parameters measured in silico from VIP, SST, and PV GABAergic neurons with varying levels of KCNT1 GOF (and, in some cases, $I_{NaP}$). For an explanation of the parameters, see Methods. AP = action potential. For each subgroup, 10 model neurons were used and simulations were repeated with the $EC_{50}$ (in mM) for $Na^+$ for the KCNT1 conductance set at the indicated levels. '+SST' refers to the activation curve parameters ($V_{50}$ and slope) for the KCNT1 conductance in that neuron type being replaced with those measured in SST neurons. Values shown are means ± the standard error. The statistical test is a repeated measures ANOVA.

• MDAR checklist

## Data availability

All data generated or analyzed during this study are included in the manuscript and supporting files; source data files have been provided for all figures.

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
