## [Editor Report · eLife assessment]

Shore et al. report the **important** effects of a heterozygous mutation in the KCNT1 potassium channel on ion currents and firing behavior of excitatory and inhibitory neurons in the cortex of KCNT1-Y777H mice. The authors provide **solid** evidence of physiological differences between this heterozygous mutation and their previous work with homozygotes. The reviewers appreciated the inclusion of recordings in ex vivo slices and dissociated cortical neurons, as well as the additional evidence showing an increase in persistent sodium currents in parvalbumin-positive interneurons in heterozygotes.

---

## [Referee Report · Reviewer #1 (Public Review)]

Summary:

This manuscript reports the effects of a heterozygous mutation in the KCNT1 potassium channels on the properties of ion currents and firing behavior of excitatory and inhibitory neurons in the cortex of mice expressing KCNT1-Y777H. In humans, this mutation as well as multiple other heterozygotic mutations produce very severe early-onset seizures and produce a major disruption of all intellectual function. In contrast, in mice, this heterozygous mutation appears to have no behavioral phenotype or any increased propensity to seizures. A relevant phenotype is, however, evident in mice with the homozygous mutation, and the authors have previously published the results of similar experiments with the homozygotes. As perhaps expected, the neuronal effects of the heterozygous mutation presented in this manuscript are generally similar but markedly smaller than the previously published findings on homozygotes. There are, however, some interesting differences, particularly on PV+ interneurons, which appear to be more excitable than wild type in the heterozygotes but more excitable in the heterozygotes. This raises the interesting question, which has been explicitly discussed by the authors in the revised manuscript, as to whether the reported changes represent homeostatic events that suppress the seizure phenotype in the mouse heterozygotes or simply changes in excitability that do not reach the threshold for behavioral outcomes.

---

## [Referee Report · Reviewer #2 (Public Review)]

Summary:

In this manuscript, Shore et al. investigate the consequent changes in excitability and synaptic efficacy of diverse neuronal populations in an animal model of juvenile epilepsy. Using electrophysiological patch-clamp recordings from dissociated neuronal cultures, the authors find diverging changes in two major populations of inhibitory cell types, namely somatostatin (SST)- and parvalbumin (PV)-positive interneurons, in mice expressing a variant of the KCNT1 potassium channel. They further suggest that the differential effects are due to a compensatory increase in the persistent sodium current in PV interneurons in pharmacological and in silico experiments. It remains unclear why this current is selectively enhanced in PV-interneurons.

Strengths:

(1) Heterozygous KCNT1 gain of function variant was used which more accurately models the human disorder.

(2) The manuscript is clearly written, and the flow is easy to follow. The authors explicitly state the similarities and differences between the current findings and the previously published results in the homozygous KCNT1 gain of function variant.

(3) This study uses a variety of approaches including patch clamp recording, in silico modeling and pharmacology that together make the claims stronger.

(4) Pharmacological experiments are fraught with off-target effects and thus it bolsters the authors' claims when multiple channel blockers (TTX and VU170) are used to reconstruct the sodium-activated potassium current.

---

## [Referee Report · Reviewer #3 (Public Review)]

Summary:

The present manuscript by Shore et al. entitled Reduced GABAergic Neuron Excitability, Altered Synaptic Connectivity, and Seizures in a KCNT1 Gain-of-Function Mouse Model of Childhood Epilepsy" describes in vitro and in silico results obtained in cortical neurons from mice carrying the KCNT1-Y777H gain-of-function (GOF) variant in the KCNT1 gene encoding for a subunit of the Na+-activated K+ (KNa) channel. This variant corresponds to the human Y796H variant found in a family with Autosomal Dominant Nocturnal Frontal lobe epilepsy. The occurrence of GOF variants in potassium channel encoding genes is well known, and among potential pathophysiological mechanisms, impaired inhibition has been documented as responsible for KCNT1-related DEEs. Therefore, building on a previous study by the same group performed in homozygous KI animals, and considering that the largest majority of pathogenic KCNT1 variants in humans occur in heterozygosis, the Authors have investigated the effects of heterozygous Kcnt1-Y777H expression on KNa currents and neuronal physiology among cortical glutamatergic and the 3 main classes of GABAergic neurons, namely those expressing vasoactive intestinal polypeptide (VIP), somatostatin (SST), and parvalbumin (PV), crossing KCNT1-Y777H mice with PV-, SST- and PV-cre mouse lines, and recording from GABAergic neurons identified by their expression of mCherry (but negative for GFP used to mark excitatory neurons).

The results obtained revealed heterogeneous effects of the variant on KNa and action potential firing rates in distinct neuronal subpopulations, ranging from no change (glutamatergic and VIP GABAergic) to decreased excitability (SST GABAergic) to increased excitability (PV GABAergic). In particular, modelling and in vitro data revealed that an increase in persistent Na current occurring in PV neurons was sufficient to overcome the effects of KCNT1 GOF and cause an overall increase in AP generation.

Strengths:

The paper is very well written, the results clearly presented and interpreted, and the discussion focuses on the most relevant points.

The recordings performed in distinct neuronal subpopulations both in primary neuronal cultures and, for some subpopulations, in cortical slices, are a clear strength of the paper. The finding that the same variant can cause opposite effects and trigger specific homeostatic mechanisms in distinct neuronal populations is very relevant for the field, as it narrows the existing gap between experimental models and clinical evidence.

---

## [Author Response]

The following is the authors’ response to the previous reviews.

**eLife assessment**
Shore et al. report important effects of a heterozygous mutation in the KCNT1 potassium channel on ion currents and firing behavior of excitatory and inhibitory neurons in the cortex of KCNT1-Y777H mice. The authors provide solid evidence of physiological differences between this heterozygous mutation and their previous work with homozygotes. The reviewers appreciated the inclusion of recordings in ex vivo slices and dissociated cortical neurons, as well as the additional evidence showing an increase in persistent sodium currents (INaP) in parvalbumin-positive interneurons in heterozygotes. However, they were unclear regarding the likelihood of the increased sodium influx through INaP channels increasing sodium-activated potassium currents in these neurons.

Regarding the last sentence of the eLife assessment, we’ve added a new paragraph to the Discussion section of the paper to address this concern. Please see the response to comment 1B of Reviewer #1 below for more details. We feel that the question of whether an increase in INaP would further increase KCNT1 activity is a valid discussion point but not a limitation of the importance or rigor of the work itself.

**Public Reviews:**

**Reviewer #1 (Public Review):**
Summary:This manuscript reports the effects of a heterozygous mutation in the KCNT1 potassium channels on the properties of ion currents and firing behavior of excitatory and inhibitory neurons in the cortex of mice expressing KCNT1-Y777H. In humans, this mutation as well as multiple other heterozygotic mutations produce very severe early-onset seizures and produce a major disruption of all intellectual function. In contrast, in mice, this heterozygous mutation appears to have no behavioral phenotype or any increased propensity to seizures. A relevant phenotype is, however, evident in mice with the homozygous mutation, and the authors have previously published the results of similar experiments with the homozygotes. As perhaps expected, the neuronal effects of the heterozygous mutation presented in this manuscript are generally similar but markedly smaller than the previously published findings on homozygotes. There are, however, some interesting differences, particularly on PV+ interneurons, which appear to be more excitable than wild type in the heterozygotes but more excitable in the heterozygotes. This raises the interesting question, which has been explicitly discussed by the authors in the revised manuscript, as to whether the reported changes represent homeostatic events that suppress the seizure phenotype in the mouse heterozygotes or simply changes in excitability that do not reach the threshold for behavioral outcomes.Strengths and Weaknesses:(1) The authors find that the heterozygous mutation in PV+ interneurons increases their excitability, a result that is opposite from their previous observation in neurons with the corresponding homozygous mutation. They propose that this results from the selective upregulation of a persistent sodium current INaP in the PV+ interneurons. These observations are very interesting ones, and they raised some issues in the original submission:A) The protocol for measuring the INaP current could potentially lead to results that could be (mis)interpreted in different ways in different cells. First, neither K currents nor Ca currents are blocked in these experiments. Instead, TTX is applied to the cells relatively rapidly (within 1 second) and the ramp protocol is applied immediately thereafter. It is stated that, at this time, Na currents and INaP are fully blocked but that any effects on Na-activated K currents are minimal. In theory this would allow the pre- to post- difference current to represent a relatively uncontaminated INaP. This would, however, only work if activation of KNa currents following Na entry is very slow, taking many seconds. A good deal of literature has suggested that the kinetics of activation of KNa currents by Na influx vary substantially between cell types, such that single action potentials and single excitatory synaptic events rapidly evoke KNa currents in some cell types. This is, of course, much faster than the time of TTX application. Most importantly, the kinetics of KNa activation may be different in different neuronal types, which would lead to errors that could produce different estimates of INaP in PV+ interneurons vs other cell types.In their revised manuscript, the authors have provided good data demonstrating that, at least for the PV and SST neurons, loss of KNa currents after TTX application is slow relative to the time course of loss of INaP, justifying the use of this protocol for these neuronal types.B) As the authors recognize, INaP current provides a major source of cytoplasmic sodium ions for the activation. An expected outcome of increased INaP is, therefore, further activation of KNa currents, rather than a compensatory increase in an inward current that counteracts the increase in KNa currents, as is suggested in the discussion.The authors comment in the rebuttal that, despite the fact that sodium entry through INaP is known to activate KNa channels, an increase in INaP does not necessarily imply increased KNa current. This issue should be addressed directly somewhere in the text, perhaps most appropriately in the discussion.

We’ve added the following new paragraph to the Discussion section of the manuscript to address this concern:

“As the persistent sodium current has been shown to act as a source of cytoplasmic sodium ions for KCNT1 channel activation in some neuron types (Hage & Salkoff, 2012), one might expect that the compensatory increase in INaP in YH-HET PV neurons would further increase, rather than counteract, KNa currents. Unfortunately, there is insufficient information on the relative locations of the INaP and KCNT1 channels, as well as the kinetics of sodium transfer to KCNT1 channels, among cortical neuron subtypes, and even less is known in the context of KCNT1 GOF neurons; thus, it is difficult to predict how alterations in one of these currents may affect the other. One plausible reason that increased INaP would not alter KNa currents in YH-HET PV neurons is that the particular sodium channels that are responsible for the increased INaP are not located within close proximity to the KCNT1 channels. Moreover, homeostatic mechanisms that modify the length and/or location of the sodium channel-enriched axon initial segment (AIS) in neurons in response to altered excitability are well described (Grubb & Burrone, 2010; Kuba et al., 2010); thus, it is possible that in YH-HET PV neurons, the length or location of the AIS is altered, leading to uncoupling of the sodium channels that are responsible for the increased INaP to the KCNT1 channels. Future studies will aim to further investigate potential mechanisms of neuron-type-specific alterations in NaP and KNa currents downstream of KCNT1 GOF.”

C) The numerical simulations, in general, provide a very useful way to evaluate the significance of experimental findings. Nevertheless, while the in-silico modeling suggests that increases in INaP can increase firing rate in models of PV+ neurons, there is as yet insufficient information on the relative locations of the INaP channels and the kinetics of sodium transfer to KNa channels to evaluate the validity of this specific model.The authors have now put in all of the appropriate caveats on this very nicely in the revised manuscript.(2) The effects of the KCNT1 channel blocker VU170 on potassium currents are somewhat larger and different from those of TTX, suggesting that additional sources of sodium may contribute to activating KCNT1, as suggested by the authors. Because VU170 is, however, a novel pharmacological agent, it may be appropriate to make more careful statements on this. While the original published description of this compound reported no effect on a variety of other channels, there are many that were not tested, including Na and cation channels that are known to activate KCNT1, raising the possibility of off-target effects.In the revised version, the authors have added more to the manuscript on this issue and have added a very clear discussion of this to the text (in the discussion section).This is a very clear and thorough piece of work, and the authors are to be congratulated on this. My one remaining suggestion would be to make an explicit statement about whether increased sodium influx through INaP channels, which is thought to activate KNa channels, would be likely to increase KNa current in these neurons (see comment 1B).

Please see response to comment 1B.

**Reviewer #2 (Public Review):**
Summary:In this manuscript, Shore et al. investigate the consequent changes in excitability and synaptic efficacy of diverse neuronal populations in an animal model of juvenile epilepsy. Using electrophysiological patch-clamp recordings from dissociated neuronal cultures, the authors find diverging changes in two major populations of inhibitory cell types, namely somatostatin (SST)- and parvalbumin (PV)-positive interneurons, in mice expressing a variant of the KCNT1 potassium channel. They further suggest that the differential effects are due to a compensatory increase in the persistent sodium current in PV interneurons in pharmacological and in silico experiments. It remains unclear why this current is selectively enhanced in PV-interneurons.Strengths:(1) Heterozygous KCNT1 gain of function variant was used which more accurately models the human disorder.(2) The manuscript is clearly written, and the flow is easy to follow. The authors explicitly state the similarities and differences between the current findings and the previously published results in the homozygous KCNT1 gain of function variant.(3) This study uses a variety of approaches including patch clamp recording, in silico modeling and pharmacology that together make the claims stronger.(4) Pharmacological experiments are fraught with off-target effects and thus it bolsters the authors' claims when multiple channel blockers (TTX and VU170) are used to reconstruct the sodium-activated potassium current.Weaknesses:(1) This study mostly relies on recordings in dissociated cortical neurons. Although specific WT interneurons showed intrinsic membrane properties like those reported for acute brain slices, it is unclear whether the same will be true for those cells expressing KCNT1 variants, especially when the excitability changes are thought to arise from homeostatic compensatory mechanisms. The authors do confirm that mutant SST-interneurons are hypoexcitable using an ex vivo slice preparation which is consistent with work for other KCTN1 gain of function variants (e.g. Gertler et al., 2022). However, the key missing evidence is the excitability state of mutant PV-interneurons, given the discrepant result of reduced excitability of PV cells reported by Gertler et al in acute hippocampal slices.
**Reviewer #3 (Public Review):**
Summary:The present manuscript by Shore et al. entitled Reduced GABAergic Neuron Excitability, Altered Synaptic Connectivity, and Seizures in a KCNT1 Gain-of-Function Mouse Model of Childhood Epilepsy" describes in vitro and in silico results obtained in cortical neurons from mice carrying the KCNT1-Y777H gain-of-function (GOF) variant in the KCNT1 gene encoding for a subunit of the Na+-activated K+ (KNa) channel. This variant corresponds to the human Y796H variant found in a family with Autosomal Dominant Nocturnal Frontal lobe epilepsy. The occurrence of GOF variants in potassium channel encoding genes is well known, and among potential pathophysiological mechanisms, impaired inhibition has been documented as responsible for KCNT1-related DEEs. Therefore, building on a previous study by the same group performed in homozygous KI animals, and considering that the largest majority of pathogenic KCNT1 variants in humans occur in heterozygosis, the Authors have investigated the effects of heterozygous Kcnt1-Y777H expression on KNa currents and neuronal physiology among cortical glutamatergic and the 3 main classes of GABAergic neurons, namely those expressing vasoactive intestinal polypeptide (VIP), somatostatin (SST), and parvalbumin (PV), crossing KCNT1-Y777H mice with PV-, SST- and PV-cre mouse lines, and recording from GABAergic neurons identified by their expression of mCherry (but negative for GFP used to mark excitatory neurons).The results obtained revealed heterogeneous effects of the variant on KNa and action potential firing rates in distinct neuronal subpopulations, ranging from no change (glutamatergic and VIP GABAergic) to decreased excitability (SST GABAergic) to increased excitability (PV GABAergic). In particular, modelling and in vitro data revealed that an increase in persistent Na current occurring in PV neurons was sufficient to overcome the effects of KCNT1 GOF and cause an overall increase in AP generation.Strengths:The paper is very well written, the results clearly presented and interpreted, and the discussion focuses on the most relevant points.The recordings performed in distinct neuronal subpopulations both in primary neuronal cultures and, for some subpopulations, in cortical slices, are a clear strength of the paper. The finding that the same variant can cause opposite effects and trigger specific homeostatic mechanisms in distinct neuronal populations is very relevant for the field, as it narrows the existing gap between experimental models and clinical evidence.Weaknesses:My main concern regarding the epileptic phenotype of the heterozygous mice investigated has been clarified in the revision, where the infrequent occurrence of seizures is more clearly stated. Also, a more detailed statistical analysis of the modeled neurons has been added in the revision.
**Recommendations for the authors:**

**Reviewer #1 (Recommendations For The Authors):**
This is a very clear and thorough piece of work, and the authors are to be congratulated on this. My one remaining suggestion would be to make an explicit statement about whether increased sodium influx through INaP channels, which is thought to activate KNa channels, would be likely to increase KNa current in these neurons (see comment 1B).

Please see response to comment 1B.

**Reviewer #2 (Recommendations For The Authors):**
This revised manuscript is significantly improved and addresses most of my concerns. However, I would still recommend including the ex vivo slice recordings in mutant PV-interneurons as the authors proposed in their rebuttal. The I-V recordings using sequential TTX and VU170 blockade in WT SST and PV-interneurons that are provided in the rebuttal are interesting and may point to a preferential expression of persistent sodium currents in PV-interneurons normally. It would be helpful to readers as a supplemental figure.

As proposed in the rebuttal, we are currently recording PV neurons using ex vivo slice preparations from WT and Kcnt1-YH Het mice. We look forward to including those data in a future manuscript.

We agree with the reviewer that the differences in INaP between WT PV and SST neurons are notable. The data provided in the rebuttal were only from 5 neurons/group, and they were meant to illustrate a side-by-side comparison of TTX and VU170 subtraction methods to assess KNa currents. However, in Figure 7 of the manuscript, we performed more robust measurements of INaP and observed differences in the current between WT PV and SST neurons. Thus, we’ve added the following sentence to the Results section:

“Interestingly, the mean peak amplitude of INaP in WT PV neurons was 70% larger than that in WT SST neurons (-1.42 ± 0.16 vs. -0.85 ± 0.07 pA/pF; Fig. 7B and 7D), suggesting there may be differences in sodium channel expression, localization, or regulation inherent to each neuron type that confer their differential response to KCNT1 GOF.”

References

Grubb, M. S., & Burrone, J. (2010). Activity-dependent relocation of the axon initial segment fine-tunes neuronal excitability. Nature, 465(7301), 1070-1074. https://doi.org/10.1038/nature09160

Hage, T. A., & Salkoff, L. (2012). Sodium-activated potassium channels are functionally coupled to persistent sodium currents. J Neurosci, 32(8), 2714-2721. https://doi.org/10.1523/JNEUROSCI.5088-11.2012

Kuba, H., Oichi, Y., & Ohmori, H. (2010). Presynaptic activity regulates Na(+) channel distribution at the axon initial segment. Nature, 465(7301), 1075-1078. https://doi.org/10.1038/nature09087